# Deploying green hydrogen to decarbonize China's coal chemical sector

Yang Guo [1,4] ✉, Liqun Peng [1], Jinping Tian [2] & Denise L. Mauzerall [1,3,4] ✉

China's coal chemical sector uses coal as both a fuel and feedstock and its increasing greenhouse gas (GHG) emissions are hard to abate by electrification alone. Here we explore the GHG mitigation potential and costs for onsite deployment of green $H_2$ and $O_2$ in China's coal chemical sector, using a life-cycle assessment and techno-economic analyses. We estimate that China's coal chemical production resulted in GHG emissions of 1.1 gigaton $CO_2$ equivalent ($GtCO_2eq$) in 2020, equal to 9% of national emissions. We project GHG emissions from China's coal chemical production in 2030 to be 1.3 $GtCO_2eq$, ~50% of which can be reduced by using solar or wind power-based electrolytic $H_2$ and $O_2$ to replace coal-based $H_2$ and air separation-based $O_2$ at a cost of 10 or 153 Chinese Yuan (CNY)/$tCO_2eq$, respectively. We suggest that provincial regions determine whether to use solar or wind power for water electrolysis based on lowest cost options, which collectively reduce 53% of the 2030 baseline GHG emissions at a cost of 9 CNY/$tCO_2eq$. Inner Mongolia, Shaanxi, Ningxia, and Xinjiang collectively account for 52% of total GHG mitigation with net cost reductions. These regions are well suited for pilot policies to advance demonstration projects.

Coal is both a fuel and a feedstock. Coal combusted in power plants accounted for 61% of China's coal consumption in 2020[1]. However, the share of coal used as a fuel is expected to decrease as coal power plants reduce capacity factors and become flexible power sources in order to integrate renewables[2]. Simultaneously, China's coal chemical sector has been rapidly expanding and is expected to continue to grow over the next decade[3–5]. Coal used in the coal chemical sector accounted for 24% of China's coal consumption in 2019. This percentage is expected to increase due to growing downstream demands for coal chemical products and energy security concerns around availability of oil and natural gas[6]. Greenhouse gas (GHG) emissions from the coal chemical sector are hard to abate as major $CO_2$ emissions result from chemical reactions that cannot be reduced by electrification alone. The coal chemical sector uses coal with $O_2$ and steam for coal gasification and generates CO and $H_2$ with $CO_2$ emitted as a byproduct. The water-gas shift reaction ($CO + H_2O \rightarrow CO_2 + H_2$) is then applied to increase the $H_2$/CO ratio for chemical syntheses and emits substantial $CO_2$. Onsite use

of coal for energy as well as upstream production of coal, grid electricity, and outsourced heat emit additional $CO_2$. Although reducing these emissions is necessary for climate targets, there has been little effort to decarbonize the coal chemical sector in either literature or practice.

Few studies have investigated low-carbon pathways for the coal chemical sector, including product structure adjustment, conversion efficiency improvements, and carbon capture, utilization and storage[3,6–9]. A hybrid power system integrating coal, natural gas, biomass, renewables, and nuclear was proposed as a low-carbon electricity source to produce electrolytic hydrogen for coal chemical production[10]. However, the GHG mitigation potential and costs of deploying onsite green hydrogen for coal chemical production have not been well studied to date. Here, we examine the benefits of deploying onsite renewable facilities nearby/within coal chemical plants to produce green $H_2$ and $O_2$ via water electrolysis. Such an approach replaces coal-based $H_2$ from the water-gas shift reaction and

[1]Princeton School of Public and International Affairs, Princeton University, Princeton, NJ 08544, USA. [2]School of Environment, Tsinghua University, Beijing 100084, China. [3]Department of Civil and Environmental Engineering, Princeton University, Princeton, NJ 08544, USA. [4]These authors jointly supervised this work: Yang Guo, Denise L. Mauzerall. ✉e-mail: yangguo@princeton.edu; mauzerall@princeton.edu

avoids substantial process-related $CO_2$ emissions (see Supplementary Fig. 1). In addition, green $O_2$ can substitute for $O_2$ from coal-driven air separation and thus reduce GHG emissions from onsite fuel combustion. Onsite renewable electricity can also replace grid electricity purchased by coal chemical plants, thus reducing upstream GHG emissions from fossil fuels used to power the grid.

Hard-to-abate sectors account for ~30% of global annual $CO_2$ emissions[11] and transitions in their fuels and feedstocks are required for a net-zero future[12,13]. Transitioning to a low-carbon society, including the use of green hydrogen, is a promising pathway to climate goals[14,15]. The chemical sector manufactures bulk materials fundamental to the economy and contributes about one eighth of global hard-to-abate emissions[11]. Emerging technologies, especially green $H_2$ applications, are necessary to address these emissions from carbon-intensive chemical reactions. The coal chemical sector is a promising large consumer of green $H_2$. Considering potential leakage and high $H_2$ transport costs in the near term[16], onsite industrial applications are critical to large-scale deployment of green $H_2$. China has recently released strategic plans that highlight the onsite use of $H_2$ from renewables in the near future[17]. China has also initiated a series of policies to facilitate low-carbon development of the coal chemical sector[4,5,18]. A demonstration project within a coal chemical enterprise in Ningxia has recently deployed a utility-scale photovoltaic (PV) system to produce green $H_2$ for coal-to-olefin processes[19].

Here we explore the GHG mitigation potential and costs to decarbonize China's coal chemical sector through the onsite use of renewable electricity to produce decarbonized $H_2$ and $O_2$ and displace carbon-intensive grid electricity. Onsite use of green $H_2$ in the coal chemical sector is a win-win opportunity. First, green $H_2$ can be used in the coal chemical sector for carbon-free feedstocks. Second, the coal chemical sector, which uses the most $H_2$ of any sector in China, will facilitate scale-up and cost reductions for green $H_2$ production. Our study provides implications for deploying renewables-based $H_2$, $O_2$, and electricity to produce a variety of coal-based chemicals, and projects the lowest cost options for each provincial region from now to about 2030. Our work goes beyond previous research to examine the role of green $H_2$ in decarbonizing the coal chemical sector nationally.

## Results
### GHG emissions in 2020 and 2030
We estimate total 2020 GHG emissions from China's coal chemical sector to be 1.12 (1.07–1.17) $GtCO_2$ equivalent ($CO_2$eq), equal to ~9% of China's GHG emissions (Fig. 1a). We convert $CO_2$, $CH_4$, and $N_2O$ emissions into $CO_2$eq using 100-year global warming potentials of 1, 28 and 265, respectively[20]. Onsite chemical reactions are responsible for 43% of total GHG emissions with the water-gas shift reaction emitting 33% alone. Onsite fuel combustion in captive power plants to generate heat and electricity accounts for 21% of total GHG emissions. Upstream processes account for the remaining 36% of the total, including grid electricity, heat supply, and coal mining and processing. Most coal chemical products have much larger onsite (from onsite chemical reactions and onsite fuel combustion) than upstream GHG emissions, except for coke and calcium carbide production which does not require $H_2$ (hence little onsite GHG emissions) but does require intensive heat and electricity (hence large upstream GHG emissions). Coke production (471 Mt in 2020) results in substantial upstream emissions of 215 $MtCO_2$eq mainly due to intensive coal mining and processing which emits 86% of its upstream GHG emissions. Traditional coal chemical products (including coke, calcium carbide, ammonia, and methanol) account for 79% of total GHG emissions from coal chemical production (Fig. 1a).

Onsite GHG emissions from coal chemical production are concentrated in coal-producing regions in China (Fig. 1b). Shanxi, Inner Mongolia, Shaanxi, and Xinjiang contributed 78% and 46% of China's coal and coal chemical production in 2020, respectively, and emitted 45% of total onsite GHG emissions from coal chemical production. We attribute the GHG emissions of each coal chemical product to the provincial region in which it was produced (see Supplementary Fig. 2). We find that modern coal chemical production (including oil, natural gas, olefin, and ethylene glycol) is mostly located in Northwest China, especially in western Inner Mongolia, Shaanxi, Ningxia, and Xinjiang; traditional coal chemical production is distributed across most provincial regions of China.

We project the 2030 baseline GHG emissions from China's coal chemical production to be 1.26 (1.19–1.33) $GtCO_2$eq, an increase of 12% relative to 2020 GHG emissions. Considering modern coal chemical projects are at a large scale and have a long construction duration (generally more than five years), we collect individual project data and assume the projects that are currently under construction or being planned will be operational in 2030. In contrast, given that traditional coal chemical projects have a relatively small scale and a short construction duration, we project their production in 2030 based on downstream demands of other sectors such as steel and agriculture[6]. Historical data, parameters and projections of coal chemical production are detailed in the Methods and Supplementary Tables 1–11. From 2020 to 2030, modern coal chemical production and related GHG

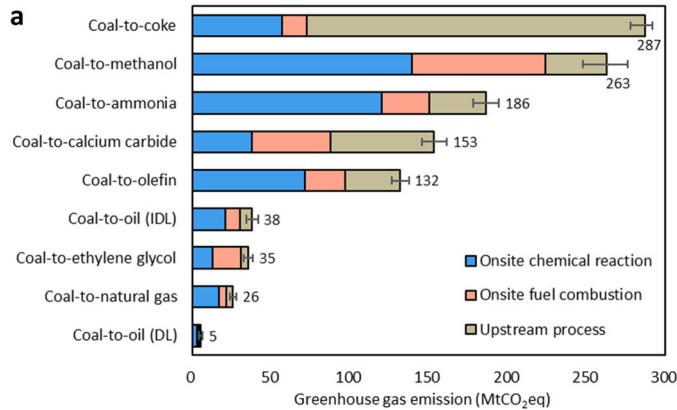

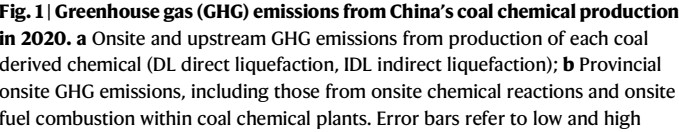

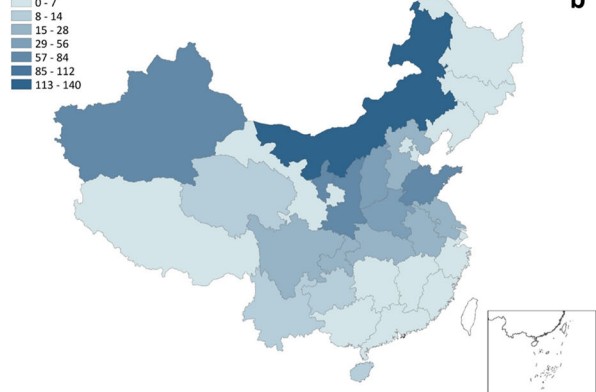

**Fig. 1 | Greenhouse gas (GHG) emissions from China's coal chemical production in 2020. a** Onsite and upstream GHG emissions from production of each coal derived chemical (DL direct liquefaction, IDL indirect liquefaction); **b** Provincial onsite GHG emissions, including those from onsite chemical reactions and onsite fuel combustion within coal chemical plants. Error bars refer to low and high estimates of GHG emissions for coal chemicals and data sources are described in the Supplementary Information. The China map is drawn by importing publicly released geographic data by the Ministry of Natural Resources of China (http://www.webmap.cn/main.do?method=index) into ArcGIS software. Source data are provided as a Source Data file.

**Table 1 | Scenario configurations for China's coal chemical sector in 2030**

| Scenarios for 2030 | Hydrogen | Oxygen | Electricity* |
|---|---|---|---|
| Baseline | Coal-based | Air separation | Moderate-RE grid |
| Moderate-renewables Grid (MG) | Electrolysis via Moderate-RE grid | | Moderate-RE grid |
| High-renewables Grid (HG) | Electrolysis via High-RE grid | | High-RE grid |
| Onsite Solar Electricity (SE) | Electrolysis via solar electricity | | Onsite solar |
| Onsite Wind Electricity (WE) | Electrolysis via wind electricity | | Onsite wind |

Electricity* refers to power that is used for water electrolysis and other operations in coal chemical plants (excluding the portion from captive coal power plants).
*RE* Renewables.

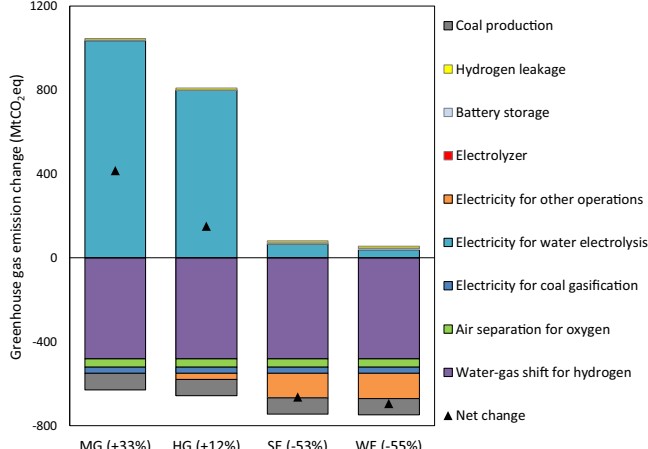

**Fig. 2 | Greenhouse gas mitigation of four alternative scenarios relative to the baseline scenario.** MG Moderate-renewables Grid scenario, HG High-renewables Grid scenario, SE Onsite Solar Electricity scenario, WE Onsite Solar Electricity scenario. Electricity for other operations refers to electricity consumption in coal chemical plants used for any process except coal gasification, air separation, or water electrolysis (such as for the water-gas shift reaction). Source data are provided as a Source Data file.

emissions are projected to rapidly increase by 113% and 93%, respectively; traditional coal chemical production and related GHG emissions, in contrast, are projected to slightly decrease by 14% and 9%, respectively. We present projected 2030 production quantities and baseline GHG emissions of coal chemicals in Supplementary Figs. 3–4.

## GHG mitigation potential in 2030

We configure a baseline scenario and four alternative scenarios for 2030 to examine the GHG mitigation potential and costs for onsite applications of electrolytic $H_2$ and $O_2$ in China's coal chemical sector, as shown in Table 1. The 2030 baseline scenario has the same industrial configurations of coal chemical production as those in 2020 ($H_2$ from coal gasification and the water-gas shift reaction, and $O_2$ from coal-driven air separation), except that grid electricity is partially decarbonized (i.e. 2030 Moderate power grid rather than 2020 power grid). The 2020 grid electricity is generated by: coal (61%), hydro (18%), wind (6%), solar (3%), nuclear (5%), gas (3%), and biomass and others (4%)[21], with a life-cycle GHG emission factor of 586 kg $CO_2$eq/MWh. The 2030 Moderate-renewables power grid (572 kg $CO_2$eq/MWh) decreases the coal share to 57% and increases the wind and solar shares to 7% and 5%, respectively, with other energy sources making up the rest of generation (hydro, 14%; nuclear, 8%; gas, 7%; biomass and others, 2%), projected by the International Energy Agency[22]. The 2030 High-renewables grid (441 kg $CO_2$eq/MWh) is further decarbonized relative to the 2030 Moderate-renewables grid, resulting in contributions to generation as follows: coal (43%), wind (16%), solar (9%), hydro (14%), nuclear (7%), gas (6%), and biomass and others (5%), projected by an integrated model for the power sector conducted by China's state-owned power companies[23]. The Moderate-renewables Grid (MG) and High-renewables Grid (HG) scenarios use the Moderate- and High-renewables grid electricity in 2030 for water electrolysis, respectively, to produce electrolytic $H_2$ and $O_2$ for coal chemical production. The Onsite Solar Electricity (SE) and Onsite Wind Electricity (WE) scenarios deploy onsite renewable energy facilities to produce green $H_2$ and $O_2$ using solar and wind electricity, respectively.

Our model includes the availability of solar and wind resources in each provincial region. It uses provincial solar/wind capacity factors to derive provincial GHG mitigation and provincial costs of deploying onsite renewable electricity generation, green $H_2$, and green $O_2$ in the coal chemical sector. We normalize GHG emissions and costs of manufacture, installation, and operation of solar/wind power facilities in the SE/WE (Onsite Solar Electricity / Onsite Wind Electricity) scenario over the lifetime electricity generation of each facility to obtain results per kWh generation. We consider provincial renewable energy availability by including provincial capacity factors (=annual electricity generation ÷ rated maximum electricity generation) of solar/wind power facilities in calculating their lifetime electricity generation. We thus derive GHG emissions and costs per kWh of solar/wind electricity generated for each provincial region across China (see Methods). As

coal chemical production requires continuous $H_2$ supply, our model includes battery storage to address the intermittency of solar and wind energy. Battery storage is used to provide stable electricity from renewables to continuously generate $H_2$ and $O_2$ via water electrolysis[24] for chemical syntheses. We quantify the GHG emissions and costs of battery storage for renewable electricity in the SE/WE scenario (see Methods).

We quantify the GHG emissions of four alternative scenarios and then compare them to the baseline scenario to identify their GHG mitigation potential, as shown in Fig. 2. In four alternative scenarios, electrolytic $H_2$ and $O_2$ required for coal chemical production are 31 Mt and 65 Mt, respectively (detailed in Supplementary Tables 12–17). We find that MG and HG scenarios increase GHG emissions by 33% and 12% (416 and 151 Mt$CO_2$eq), respectively, relative to the baseline scenario. This indicates that using grid electricity for water electrolysis in 2030, even if a relatively high fraction comes from renewables, is not a low-carbon option for the coal chemical sector. However, the SE and WE scenarios substantially reduce GHG emissions by 53% and 55% (664 and 694 Mt$CO_2$eq), respectively, relative to the baseline scenario. This indicates that deploying onsite renewable energy with water electrolysis to generate $H_2$, $O_2$, and electricity is a promising decarbonization pathway for China's coal chemical sector.

We further decompose GHG emission changes to individual industrial processes to identify critical contributors (Fig. 2). All four alternative scenarios significantly reduce onsite GHG emissions from chemical reactions by 482 Mt$CO_2$eq, due to the removal of the water-gas shift reaction which is a primary $CO_2$ emitter in coal chemical production systems. The four alternative scenarios also reduce onsite fuel combustion emissions by 69 Mt$CO_2$eq by use of electrolytic $O_2$ instead of air separation-based $O_2$ (−39 Mt$CO_2$eq) and reductions in onsite coal-based electricity generation for coal gasification (−30 Mt$CO_2$eq, due to avoiding CO production for the water-gas shift reaction). As a byproduct of water electrolysis, electrolytic $O_2$ can replace original $O_2$ and thus cut down onsite fuel combustion for air separation devices. As coal consumption to generate $H_2$ and to drive air separation for $O_2$ is removed, upstream GHG emissions from coal mining and processing decrease by 78 Mt$CO_2$eq in all four alternative scenarios. In the HG, SE, and WE scenarios, we also find a decrease of 28, 116, and 119 Mt$CO_2$eq, respectively, in upstream GHG emissions from grid electricity. This is due to the replacement of the Moderate-

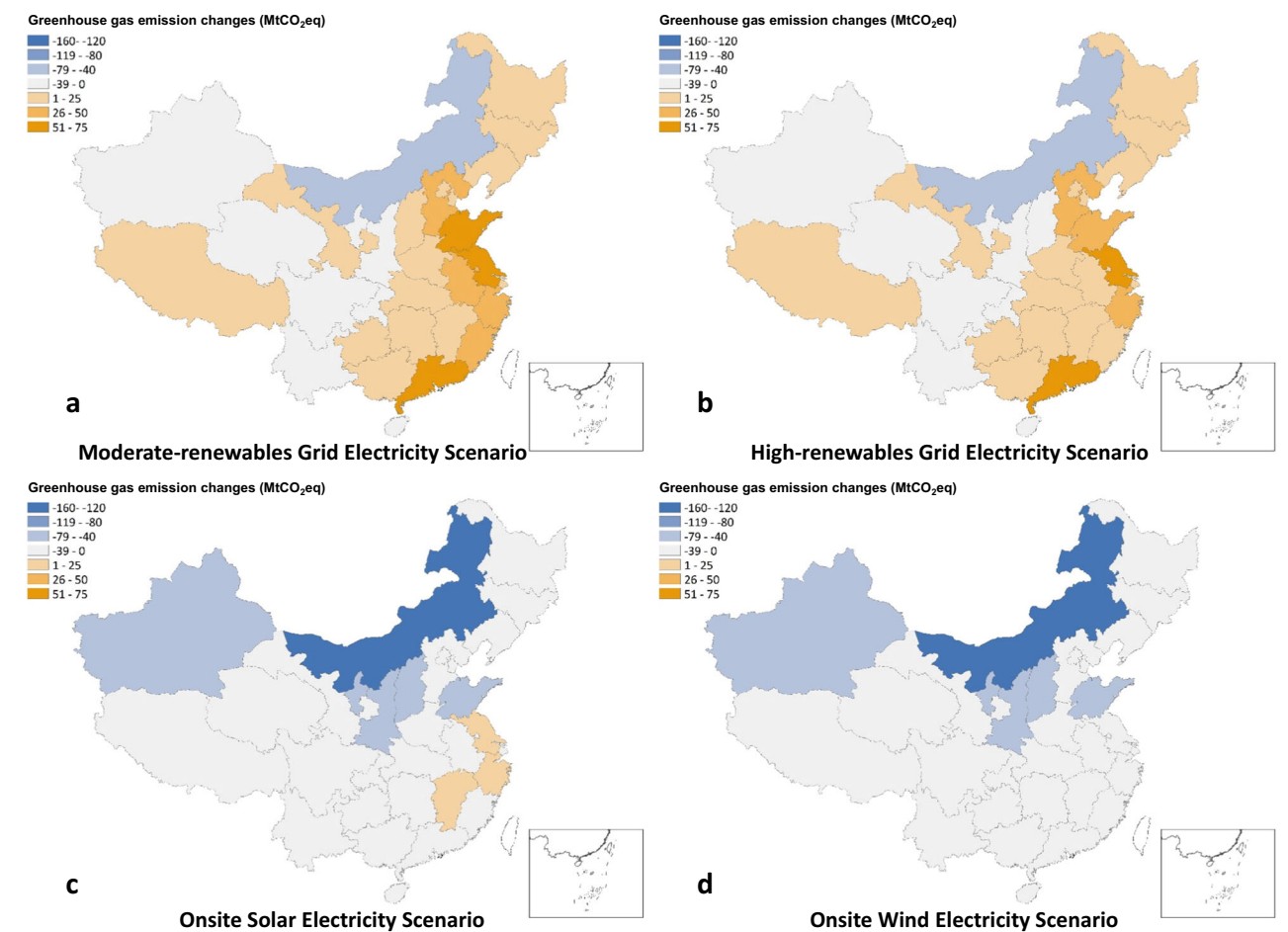

**Fig. 3 | Provincial greenhouse gas emission changes in the four alternative scenarios relative to the baseline scenario. a** Moderate-renewables Grid Electricity scenario; **b** High-renewables Grid Electricity scenario; **c** Onsite Solar Electricity scenario; and **d** Onsite Wind Electricity scenario. The China map is drawn by importing publicly released geographic data by Ministry of Natural Resources of China (http://www.webmap.cn/main.do?method=index) into ArcGIS software. Source data are provided as a Source Data file.

renewables grid electricity with the High-renewables grid electricity or onsite renewable electricity for all plant operations except water electrolysis.

However, deploying water electrolysis in coal chemical plants increases upstream GHG emissions from electricity generation for water electrolysis as well as manufacturing processes for electrolyzers, solar panels/wind turbines, and battery storage. Additional electricity for water electrolysis in MG and HG scenarios results in a significant increase of ~1000 and ~800 MtCO₂eq, respectively, in upstream GHG emissions from the power grid, and thus more than offsets GHG emission reductions of ~630 and ~660 MtCO₂eq, respectively, from other industrial processes. We estimate increases in annual GHG emission resulting from onsite renewable electricity used for water electrolysis as 65 and 38 MtCO₂eq in the SE and WE scenarios, respectively. We estimate that ~330 GW of water electrolyzers is needed in the four alternative scenarios, and ~320 GW of battery storage with a four-hour discharge rate is needed in the SE and WE scenarios. Limited increases in upstream GHG emissions result from manufacturing electrolyzers and batteries, which are 2.7 MtCO₂eq (in the four alternative scenarios) and 6.4 MtCO₂eq (in the SE and WE scenarios), respectively. We also include H₂ leakage in our analysis which adds to global warming because H₂ is an indirect GHG and extends the lifetime of CH₄ in the atmosphere. We use 100-year global warming potential of 11 for H₂[25] and quantify H₂ leakage during H₂ production to be 7.5 MtCO₂eq in the four alternative scenarios. See calculations for GHG mitigation in Supplementary Tables 14–17.

We attribute GHG emission changes in all four alternative scenarios to provincial regions where industrial processes physically occur (Fig. 3). Changes in onsite GHG emissions occur in provincial regions where coal chemicals are produced, including those from chemical reactions, onsite fuel combustion, and H₂ leakage. We allocate GHG emission changes in coal production, grid electricity generation, and solar power facility manufacturing to provincial regions based on provincial production of coal, thermal power, and solar panels, respectively (see Supplementary Table 17)[1,26]. We attribute GHG emissions from wind power facility manufacturing to provincial regions where wind turbines are deployed, since building materials and nacelles (mainly consisting of concrete and steel) account for major GHG emissions[27] and they have short cost-effective transport distances. We also assume that electrolyzer and battery manufacturing occurs in local provincial regions. Such an attribution approach for upstream environmental impacts has been applied in related studies[28,29]. We detail the attribution method in Supplementary Table 14.

In the MG and HG scenarios, most provincial regions increase their GHG emissions relative to the baseline scenario because their electricity generation increases as grid electricity demand increases for water electrolysis. Beijing and Tibet, although without coal chemical production, provide a portion of grid electricity for grid-based water electrolysis occurring in other provincial regions in the MG and HG scenarios, which increases their GHG emissions relative to the baseline scenario. However, some coal-rich regions such as Inner

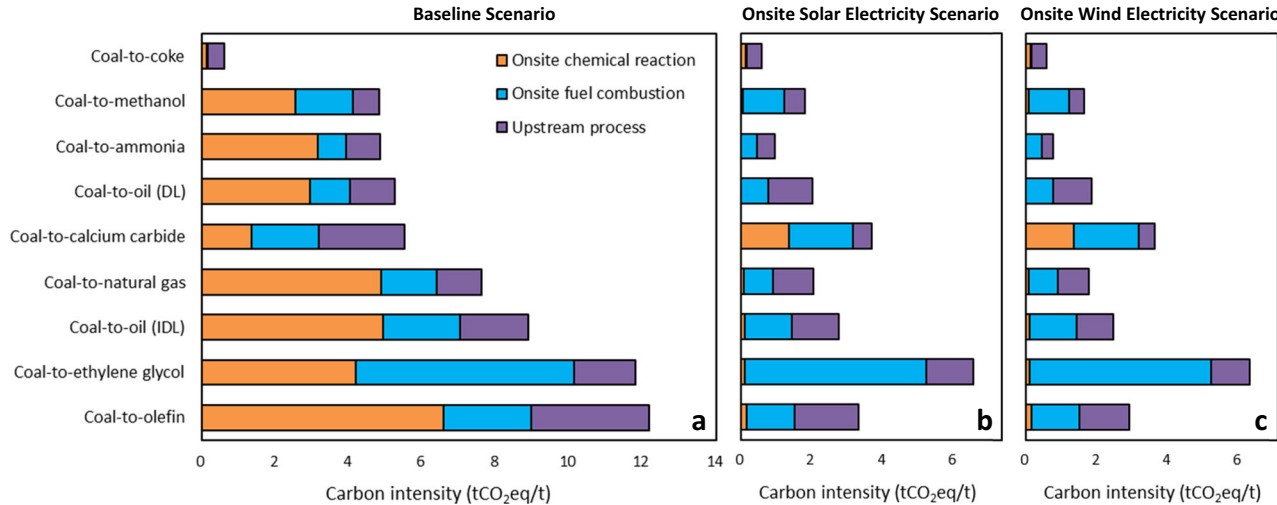

**Fig. 4 | Carbon intensities of coal chemicals in the baseline, Onsite Solar Electricity, and Onsite Wind Electricity scenarios. a** Baseline scenario; **b** Onsite Solar Electricity scenario; and **c** Onsite Wind Electricity scenario. DL direct liquefaction, IDL indirect liquefaction. Source data are provided as a Source Data file.

Mongolia decrease their GHG emissions due to substantial reductions in GHG emissions from onsite coal chemical reactions and coal mining. In addition, Sichuan, Chongqing, Yunnan, and Hainan also reduce their GHG emissions because they reduce onsite coal chemical production emissions while only slightly increasing grid electricity generation emissions due to a low fraction of thermal power in their power mix. In the SE and WE scenarios, regions with massive coal chemical production have the most GHG mitigation compared with the baseline scenario due to substantial onsite GHG emission reductions from chemical reactions and fuel combustion. Especially, Inner Mongolia, Xinjiang, and Shaanxi decrease their GHG emissions by ~140, ~75, and ~70 MtCO$_2$eq, respectively in the SE and WE scenarios. However, Jiangsu, Zhejiang, and Jiangxi slightly increase their GHG emissions in the SE scenario by 5.6, 0.5, and 2.5 MtCO$_2$eq, respectively, when using solar-based H$_2$ and O$_2$. This is because the three regions produce the majority (~60%) of solar panels in China which results in additional GHG emissions. Beijing also provides solar panels for solar-based water electrolysis occurring in other regions in the SE scenario, but its avoided emissions from reductions in grid electricity generation more than offset those from its solar panel production. Therefore, Beijing decreases GHG emissions in the SE scenario relative to the baseline scenario.

We further conduct a comparative analysis of carbon emission intensities of coal chemical products in the 2030 baseline, SE, and WE scenarios (Fig. 4 and Supplementary Table 18). Overall, the SE and WE scenarios both reduce the carbon intensities by about 35–85% for various coal chemicals except for coke (only 5%), relative to the baseline carbon scenario. The two scenarios substantially reduce carbon intensities of onsite chemical reactions except for coke and calcium carbide production which requires no H$_2$ and O$_2$. As there are energy savings resulting from the removal of air separation for O$_2$ production and reductions in coal gasification for CO production, carbon intensities of onsite fuel combustion in the SE and WE scenarios decrease by 14–44% for various coal chemicals. Captive power plants in coal chemical enterprises combust coal to supply heat for chemical reaction systems and to also generate electricity for balancing renewable electricity used in water electrolysis. Carbon intensities of upstream processes in the SE and WE scenarios also largely decrease due to using renewable electricity instead of grid electricity and reducing coal mining and processing. Carbon intensities of coal chemicals in the WE scenario are slightly lower than in the SE scenario due to a lower national average GHG emission factor of wind electricity generation (20 kgCO$_2$eq/MWh[30]) than solar (36 kgCO$_2$eq/MWh[31]).

We then analyze the remaining GHG emissions in the SE and WE scenarios (see Supplementary Fig. 5 and Supplementary Table 19). We find that GHG emissions from coke and calcium carbide production account for ~60% of the total remaining emissions in both scenarios. Further reductions in process-related GHG emissions from calcium carbide and coke production necessitate the onsite deployment of carbon capture, utilization, and storage. Onsite fuel combustion emissions (~40%) and upstream emissions (~45%) collectively account for ~85% of total remaining emissions in both scenarios. Onsite fuel combustion emissions can be further reduced by replacing captive coal power plants with heat and electricity supplied from a decarbonized power grid.

## GHG mitigation costs in 2030

We quantify GHG mitigation costs in 2030 by identifying cost changes in renewable energy scenarios (SE and WE) relative to the baseline scenario, as shown in Table 2. We use the 2030 prices of energy and equipment for such cost analyses. Annual costs in coal consumption as a feedstock and fuel are reduced by 290 billion CNY in the SE and WE scenarios. Electrolyzers and battery storage annually add 120 and 87 billion CNY in costs, respectively. Additional renewable electricity for water electrolysis increases annual costs by 193 and 282 billion CNY in the SE and WE scenarios, respectively; costs of electricity consumption for other plant operations are reduced by 103 and 92 billion CNY annually in the SE and WE scenarios, respectively, due to replacement of grid electricity with onsite renewable electricity. Therefore, net costs for GHG mitigation are 6 and 106 billion CNY, respectively, in the SE and WE scenarios. Given that SE and WE reduce GHG emissions by 664 and 694 MtCO$_2$eq, respectively, unit costs for GHG mitigation are 10 and 153 CNY per ton of CO$_2$eq. We derive low and high estimates of cost changes due to uncertainty of energy and equipment prices as well as various discount rates used for accounting (detailed in Supplementary Tables 20–21). We find that coal cost reductions can more than offset equipment cost additions in some cases of the SE and WE scenarios, thus deriving net economic benefits. Uncertainty ranges in Table 2 indicate that total net costs in the SE and WE scenarios are sensitive to price variations in coal for feedstocks and renewable energy generation.

We attribute national cost changes to provincial regions to analyze the geographic heterogeneity of GHG mitigation costs for the coal chemical sector (Fig. 5). We find that renewables-rich regions such as Xinjiang, Ningxia and Inner Mongolia have small cost additions or even cost reductions when using renewables-based H$_2$ and O$_2$ for coal

**Table 2 | GHG mitigation costs in 2030 for China's coal chemical sector relative to the baseline scenario (in 2020 Chinese Yuan)**

| Indicator | Annual cost change (billion CNY) | |
| --- | --- | --- |
| | Onsite Solar Electricity scenario | Onsite Wind Electricity scenario |
| Coal for feedstocks | −267 (−331 to −203) | −267 (−331 to −203) |
| Coal for fuels | −23 (−27 to −20) | −23 (−27 to −20) |
| Electrolyzer | 120 (114 to 123) | 120 (114 to 123) |
| Battery | 87 (85 to 89) | 87 (85 to 89) |
| Electricity for water electrolysis | 193 (184 to 198) | 282 (273 to 287) |
| Electricity for other operations | −103 (−104 to −102) | −92 (−93 to −92) |
| Net change | 6.4 (−79 to 84) | 106 (21 to 184) |
| CNY per tCO₂eq | 10 (−120 to 127) | 153 (30 to 265) |

"+/−" refers to increases/decreases in costs. Numbers in parentheses refer to low and high estimates. Cost additions of renewable electricity, battery storage, and electrolyzers include both capital and operating costs. Capital costs of renewable energy facilities, battery storage, and water electrolyzers are levelized based on lifetime electricity generation or lifetime hydrogen production, i.e., CNY/MWh or CNY/tH₂. Electricity for other operations refers to electricity consumption in coal chemical plants used for any process except coal gasification, air separation, or water electrolysis (such as for the water-gas shift reaction). We use the 2030 prices of energy and equipment for cost analyses. Parameters and estimates are described in Supplementary Tables 20–21. *CNY* Chinese Yuan.

chemical production. For a given region, cost changes in the SE and WE scenarios can be quite different such as Ningxia and Inner Mongolia. Provincial regions may determine whether to use solar or wind-based water electrolysis based on their renewable resources to reduce costs.

GHG mitigation costs in the SE and WE scenarios are equal to 0.6% and 9%, respectively, of total costs for coal chemical production in 2030 baseline scenario (-1100 billion CNY, see Supplementary Table 22). Cost additions (CNY/t) for coal chemicals except olefin, coke, and calcium carbide in the SE and WE scenarios are equal to 11–33% and 34–61%, respectively, of their current national average production costs (Supplementary Table 22). This necessitates the inclusion of the coal chemical sector in China's carbon trading market to mitigate the cost burden of coal chemical enterprises when they deploy renewables-based H₂ and O₂ for decarbonization. However, these two scenarios reduce the costs of coal-to-coke and coal-to-calcium carbide by about 2% and 60%, respectively, of their current national average production costs due to using onsite renewable electricity (national average of 109 CNY/MWh for solar and 155 CNY/MWh for wind, see Supplementary Table 21) to replace costly grid electricity (580 CNY/MWh, see Supplementary Table 20). For olefin, the SE scenario decreases the olefin production costs by 1%, while the WE scenario increases the costs by 19%. Overall, solar-based H₂ for chemical syntheses delivers more cost-competitive chemical products than wind-based.

| | Coal for feedstocks | Coal for fuels | Electrolyzer | Battery | Electricity_SE | Total_SE | Electricity_WE | Total_WE |
| --- | --- | --- | --- | --- | --- | --- | --- | --- |
| Beijing | | | | | | | | |
| Tianjin | -0.60 | -0.050 | 0.27 | 0.20 | 0.34 | 0.15 | 0.40 | 0.21 |
| Hebei | -5.5 | -0.44 | 2.5 | 1.8 | 2.0 | 0.3 | 2.6 | 0.9 |
| Shanxi | -14 | -1.2 | 6.5 | 4.7 | 4.6 | 0.2 | 11 | 6.5 |
| Inner Mongolia | -57 | -5.2 | 25 | 19 | 11 | -7.4 | 46 | 28 |
| Liaoning | -8.3 | -0.69 | 3.7 | 2.7 | 5.2 | 2.7 | 7.0 | 4.5 |
| Jilin | -0.65 | -0.044 | 0.29 | 0.21 | 0.28 | 0.09 | 0.27 | 0.08 |
| Heilongjiang | -1.4 | -0.11 | 0.65 | 0.47 | 0.65 | 0.21 | 0.63 | 0.19 |
| Shanghai | -1.3 | -0.13 | 0.59 | 0.43 | 0.85 | 0.43 | 0.74 | 0.32 |
| Jiangsu | -5.8 | -0.42 | 2.6 | 1.9 | 3.2 | 1.5 | 6.3 | 4.5 |
| Zhejiang | -1.2 | -0.089 | 0.55 | 0.40 | 0.68 | 0.31 | 0.75 | 0.38 |
| Anhui | -8.8 | -0.73 | 3.9 | 2.9 | 4.7 | 2.0 | 9.6 | 6.9 |
| Fujian | -1.6 | -0.12 | 0.7 | 0.52 | 0.78 | 0.31 | 1.1 | 0.58 |
| Jiangxi | -0.14 | -0.010 | 0.064 | 0.047 | -0.50 | -0.54 | -0.36 | -0.41 |
| Shandong | -20 | -1.7 | 9 | 6.7 | 12 | 6 | 14 | 7.8 |
| Henan | -16 | -1.3 | 7 | 5.3 | 8.9 | 4.0 | 25 | 20 |
| Hubei | -7.7 | -0.56 | 3.5 | 2.5 | 4.2 | 1.9 | 6.1 | 3.8 |
| Hunan | -1.5 | -0.10 | 0.67 | 0.49 | 0.65 | 0.22 | 1.1 | 0.68 |
| Guangdong | -0.010 | -0.00064 | 0.0043 | 0.0031 | -0.13 | -0.13 | -0.11 | -0.11 |
| Guangxi | -1.6 | -0.11 | 0.7 | 0.52 | 0.69 | 0.22 | 1.7 | 1.2 |
| Hainan | -3.1 | -0.27 | 1.4 | 1.0 | 1.6 | 0.6 | 2.2 | 1.3 |
| Chongqing | -5.6 | -0.46 | 2.5 | 1.8 | 4.0 | 2.3 | 5.2 | 3.5 |
| Sichuan | -5.7 | -0.40 | 2.6 | 1.9 | 1.0 | -0.68 | 1.6 | -0.07 |
| Guizhou | -4.4 | -0.35 | 2.0 | 1.5 | 2.4 | 1.0 | 3.7 | 2.3 |
| Yunnan | -3.9 | -0.28 | 1.8 | 1.3 | 0.18 | -0.99 | 1.8 | 0.61 |
| Tibet | | | | | | | | |
| Shaanxi | -27 | -2.6 | 12 | 8.8 | 7.8 | -0.9 | 12 | 3.0 |
| Gansu | -1.2 | -0.094 | 0.54 | 0.39 | -1.0 | -1.3 | -0.14 | -0.50 |
| Qinghai | -5.7 | -0.55 | 2.5 | 1.9 | 0.83 | -0.99 | 5.1 | 3.3 |
| Ningxia | -26 | -2.5 | 12 | 8.6 | 4.4 | -4.0 | 12 | 3.3 |
| Xinjiang | -31 | -2.7 | 14 | 10 | 8.9 | -0.7 | 13 | 2.9 |

-57 ▓▓▓▓▓░░░░▓▓▓▓▓ 46 (billion Chinese Yuan)

**Fig. 5 | Provincial greenhouse gas mitigation costs for China's coal chemical sector in 2030 relative to the baseline scenario (in 2020 Chinese Yuan).** SE Onsite Solar Electricity scenario, WE Onsite Wind Electricity scenario. Source data are provided as a Source Data file.

## Discussion

The coal chemical sector is a growing carbon emitter in China and its GHG emissions are hard to abate by electrification alone. Few previous studies have explored decarbonization strategies for China's coal chemical sector and characterized their GHG mitigation potential and costs. Here we examine onsite deployment of green $H_2$ and $O_2$ in coal chemical plants, one of the most promising decarbonization measures in this hard-to-abate industrial sector. We demonstrate that even using grid electricity for water electrolysis with ~50% of generation derived from renewables and nuclear power and ~50% from fossil fuels in 2030 is not a low-carbon option for the coal chemical sector and will increase GHG emissions by 12% relative to the baseline scenario while also increasing costs by ~880 billion CNY annually (Supplementary Table 20). In contrast, onsite deployment of renewables-based electrolytic $H_2$ and $O_2$ to replace coal-based $H_2$ and air separation-based $O_2$ in the coal chemical sector is more effective at reducing GHG emissions with lower costs. We find that using solar and wind-based hydrogen, oxygen, and electricity can reduce 53% and 55%, respectively, of 2030 baseline GHG emissions from coal chemical production in China. Since onsite renewable electricity (national average of 109 CNY/MWh for solar and 155 CNY/MWh for wind, see Supplementary Table 21) in 2030 is cheaper than grid electricity (580 CNY/MWh, see Supplementary Table 20), GHG mitigation costs when using renewables-based water electrolysis are much lower than when using grid electricity-based. However, onsite solar and wind-based water electrolysis will increase coal chemical production costs by 6 and 106 billion CNY, respectively, in 2030.

The decarbonization of China's coal chemical sector is also an opportunity to develop green hydrogen at scale, which will boost technical innovation and decrease the costs of green hydrogen production. According to China's 2022 strategic plan for green hydrogen, onsite applications in industrial sectors are especially encouraged[17]. The coal chemical sector is currently the largest producer and consumer of hydrogen in China[32]. We estimate the water-gas shift reaction in coal chemical systems produced 17 Mt coal-based $H_2$ in 2020, accounting for more than 50% of China's total $H_2$ production (33 Mt)[33]. We also project that China's coal chemical production in 2030 will require 21 Mt green $H_2$ to replace water-gas shift-based $H_2$. Coupling of coal chemical production and green hydrogen is a win-win opportunity to both scale up the deployment of green $H_2$ and to utilize low-carbon feedstocks for coal-chemical production.

We explore the best options for each provincial region to decarbonize the coal chemical sector via either solar or wind-based $H_2$ to achieve the maximum national GHG mitigation and minimum national costs. We use results from the Onsite Solar Electricity and Onsite Wind Electricity scenarios to optimize the combinations of provincial options to maximize national GHG mitigation and to minimize national costs. We assign each provincial region to either solar or wind-based water electrolysis based on which technology yields larger GHG mitigation or lower costs within each region. In the maximum national GHG mitigation solution, we find a reduction of −57% (−722 MtCO$_2$eq) relative to 2030 baseline GHG emissions at a national annual cost addition of 26 billion CNY (36 CNY/tCO$_2$eq) relative to baseline costs. In this transition, 17 provincial regions deploy solar- and 12 provincial regions deploy wind-based water electrolysis (excluding Beijing and Tibet where no coal chemicals are produced, and Hong Kong, Macau, and Taiwan where no coal chemical data are available). In the minimum national cost solution, we find a national annual cost increase of 6.3 billion CNY (9.4 CNY/tCO$_2$eq) in 2030 relative to baseline costs with national GHG mitigation of −53% (−665 MtCO$_2$eq) relative to baseline emissions. In this transition, 26 provincial regions deploy solar- and 3 provincial regions deploy wind-based water electrolysis, with the same exclusions as above. These two solutions have similar GHG mitigation (−57% vs. −53% relative to the baseline) but costs for the minimum cost solution are only 24% of those for the maximum mitigation solution.

We thus suggest that provincial regions determine whether to use onsite solar or wind power for water electrolysis based on their lowest cost options (see Supplementary Table 23). We further find that Inner Mongolia, Shaanxi, Ningxia, and Xinjiang have much larger GHG mitigation potential than other provincial regions while simultaneously achieving net cost reductions due to their abundant solar energy. These four provincial regions collectively account for 52% of total GHG mitigation in the minimum national cost solution. For policymaking and demonstration projects, the four provincial regions can be pioneers in cost-effectively deploying onsite green $H_2$ and $O_2$ in coal chemical production. This clearly shows the enormous potential of decarbonizing the coal chemical sector at only a small cost increase at the national level.

Excess $O_2$ from water electrolysis (except that used in coal gasification) can be sold to increase GHG mitigation and revenue by replacing $O_2$ from coal-driven air separation. Excess $O_2$ sales are not included in the results of GHG mitigation potential and costs. We estimate that using green $O_2$ can mitigate 0.26 tCO$_2$eq/tO$_2$ compared with $O_2$ from coal-driven air separation. If excess green $O_2$ not needed for chemical production is sold at a price of ~360 CNY/tO$_2$ (the production cost of air separation-based $O_2$)[34], revenues from 185 Mt of excess $O_2$ (250 Mt generated from water electrolysis and 65 Mt used for coal gasification) in 2030 can reach 67 billion CNY (10.5 and 0.6 times the GHG mitigation costs in the SE and WE scenarios, respectively), with ~50 MtCO$_2$eq of GHG mitigation (4% of the 2030 baseline GHG emissions). Additional $O_2$ storage and transport may slightly reduce such economic and carbon benefits. In general, excess $O_2$ sales can substantially reduce costs of renewables-based water electrolysis for coal chemical production. Calculations are detailed in Supplementary Table 24.

We suggest onsite deployment of green $H_2$ in coal chemical plants because it can avoid costly long-distance $H_2$ transport[35]. Considering battery storage is more widely used than $H_2$ storage[36], we include the GHG emissions and costs of using battery storage for renewable electricity instead of $H_2$ storage to reduce costs and $H_2$ leakage. Battery storage can help provide stable renewable electricity for water electrolysis to continuously deliver green $H_2$ for coal chemical production. In practice, coal chemical plants may need very short pipelines for $H_2$ transport within plants and small-scale $H_2$ storage as a back-up, which results in insignificant increases of GHG emissions and costs.

Co-benefits for air quality and human health result from the use of green $H_2$ and $O_2$, in addition to GHG mitigation. Using green $H_2$ and $O_2$ in chemical plants can decrease onsite coal use for both feedstocks and fuels, and thus reduce air pollutant emissions from coal gasification and combustion. Avoided premature deaths from such air quality improvements can be monetized in cost-benefit analyses[37,38] to further offset GHG mitigation costs in the SE and WE scenarios.

We analyze the land area required for onsite renewable energy deployment to power water electrolysis. Capacity additions in the SE and WE scenarios are 1.1 TW of solar power and 0.96 TW of wind power, respectively, to generate 2.0 PWh of renewable electricity to power water electrolysis and replace grid electricity used in coal chemical plants. We apply land conversion factors to estimate that 26,000 km$^2$ and 70,000 km$^2$ are required to install solar and wind power, respectively, in the SE and WE scenarios. Provincial parameters and results are listed in Supplementary Table 25. 35% of coal chemical production and 56% of needed renewable electricity in 2030 is projected to be in Northwestern China including Xinjiang, Ningxia, Qinghai, Gansu, Shaanxi, and west Inner Mongolia. About 60% of total land area needed for renewables in both SE and WE scenarios is located in these six less-populated provincial regions. Thus, land availability within/surrounding coal chemical plants should not be a constraint for deploying renewable energy facilities and water electrolyzers. Also, coal chemical plants can distribute deployment using rooftop PV and

distributed wind turbines to fully utilize space within plant areas. In practice, a pilot project (~0.5 km$^2$) has been operated since 2021 in Ningxia that deploys solar power and water electrolyzers in a coal chemical plant covering 13 km$^2$. In this case, solar-based H$_2$ production requires about 4% of total plant area. Another hybrid renewables-based hydrogen project for coal chemical production using both solar and wind power is under construction in Inner Mongolia. These pilot projects demonstrate the feasibility of onsite green hydrogen applications in coal chemical production.

China's coal chemical plants generally have onsite captive coal power plants to generate heat and electricity for chemical production, with grid electricity as a supplementary power source[7]. In this study, we use onsite renewable electricity to electrolyze the water and to replace grid electricity purchased by coal chemical plants. The onsite coal power plants remain operational as a high-temperature heat source and as an electricity source for plant operations besides water electrolysis. China's chemical sector is expected to be included in the national carbon trading market by 2035[39], and high-temperature heat generation from coal is hard to replace with renewable electricity at scale in the near future. Therefore, we propose the onsite deployment of renewable electricity for water electrolysis and for the replacement of grid electricity use in coal chemical plants during 2023–2035 but retain the onsite coal power plants for other plant operations. As electrification technologies advance over the next decade, we suggest that onsite deployment of renewable electricity should increasingly replace onsite heat and power generation from coal for industrial processes (such as air separation and coal gasification).

Onsite deployment of renewables-based electrolytic H$_2$ and O$_2$ is a feasible pathway to partially decarbonize China's coal chemical sector. We suggest that provincial regions determine whether to use onsite solar or wind power for water electrolysis based on their lowest cost options, which collectively reduce 53% of the 2030 baseline GHG emissions from coal chemical production at the low cost in 2030 of 9.4 CNY/tCO$_2$eq. We find Inner Mongolia, Shaanxi, Ningxia, and Xinjiang collectively account for 52% of total GHG mitigation that is possible with net cost reductions. These four provincial regions, which have extensive available land, can be pioneers in deploying cost-effective onsite green H$_2$ and O$_2$ in coal chemical production. Excess green O$_2$ sales can substantially reduce costs of renewables-based water electrolysis for coal chemical production. GHG mitigation costs can be offset if the coal chemical sector is included in China's carbon trading market (the carbon price was ~50 CNY/tCO$_2$ in 2021[40], which makes it highly profitable to trade carbon permits when compared to the 9.4 CNY/tCO$_2$eq cost of mitigation in 2030). Coupling chemical production with green hydrogen is a win-win opportunity to both scale up the deployment of green H$_2$ and to utilize a low-carbon feedstock for the coal-chemical sector. We plan to use plant-level operational data to extend the study of the coal chemical sector to examine the environmental co-benefits of using onsite green H$_2$ for air quality improvements and freshwater conservation. We will also consider the use of onsite renewable electricity with battery storage to replace captive coal power facilities in coal chemical plants when high-temperature heat generation from electricity is feasible at scale.

## Methods

### GHG emission accounting for China's coal chemical sector

We collect or estimate provincial production of coal chemicals in 2020, including traditional coal chemicals (coke, calcium carbide, ammonia, methanol) and modern coal chemicals (oil, natural gas, olefin, and ethylene glycol). We obtain the 2020 provincial production of coal-to-coke directly from statistics[41]. We collect the 2020 national production of coal-to-calcium carbide[42], coal-to-ammonia[41,43], and coal-to-methanol[44,45], and then use up-to-date distribution patterns of

provincial production[7,46,47] to allocate the national production to provincial regions. We collect the 2020 national production of coal-to-oil (from direct and indirect coal liquefaction), coal-to-natural gas, coal-to-olefin, and coal-to-ethylene glycol[48], and then allocate the national production to provincial regions based on provincial capacity distribution of modern coal chemical projects (totaling up capacities of current individual projects)[49]. We detail provincial production of coal chemicals and individual project information of modern coal chemicals in Supplementary Tables 1–3.

We project the 2030 national production of traditional coal chemicals based on downstream sectoral demands[6], and allocate the national production to provincial regions using the same distribution patterns of provincial production as in 2020 due to stable production distribution in recent years[7]. We project the 2030 national production of modern coal chemicals assuming that individual projects currently under construction or being planned will be operational in 2030 (Supplementary Table 3). We then use the 2030 provincial capacity distribution of modern coal chemical projects to allocate the national production to provincial regions. Detailed data are in Supplementary Tables 4–6.

In addition to coal chemical production, we collect up-to-date GHG emission factors of coal chemicals[7,50] for 2020 GHG emission estimates, including GHG emission factors of onsite chemical reactions and onsite fuel combustion (Supplementary Tables 7–9). We collect or estimate life-cycle GHG emission factors of related upstream processes including coal mining and processing, grid electricity generation, and outsourced heat generation using a localized life-cycle database for China[51,52]. We use a 2020 grid electricity mix and GHG emission factors of various electricity generation technologies[51] to calculate the life-cycle GHG emission factor of 2020 grid electricity at the national level (Supplementary Tables 9–10). We assume the 2030 baseline GHG emission factors of onsite chemical reactions, onsite fuel combustion, coal production, and outsourced heat generation to be the same as in 2020, while the life-cycle GHG emission factors of 2030 grid electricity are derived using projected 2030 grid electricity mixes (Supplementary Tables 10–11). Based on production quantities and GHG emission factors of coal chemicals, we estimate total GHG emissions from China's coal chemical production in 2020 and in a 2030 baseline scenario.

### Scenario configurations and GHG mitigation modeling

We integrate techno-economic analyses with a life-cycle assessment to systematically examine the GHG mitigation potential and costs of deploying onsite green H$_2$, green O$_2$, and renewable electricity in China's coal chemical sector. Techno-economic analyses are used to examine the technical performance and cost-effectiveness of a technical process or product[53]. A life-cycle assessment is based on a series of stages in the "cradle-to-grave" life cycle of a product or technology[54]. Accordingly, we apply a broad system boundary to include onsite and upstream processes, e.g., coal gasification, the water-gas shift, air-separation for O$_2$, water electrolysis, upstream manufacturing of battery storage, water electrolyzers, and renewable power facilities, and upstream production of coal for chemical feedstocks and fuels.

Figure 6 presents the modeling framework for GHG mitigation and cost changes in the baseline scenario and four alternative scenarios for China's coal chemical production in 2030. We estimate 2030 baseline GHG emissions using projected coal chemical production and baseline GHG emission factors in 2030. We then quantify GHG mitigation potential of four alternative scenarios by comparing their GHG emissions to the baseline emissions. We use or estimate the 2030 projections of coal chemical production, GHG emission factors, and cost parameters based on literature, which may not fully reflect the real situation in the future. Finally, we model GHG mitigation and costs using annual averages for parameterizations at the provincial level,

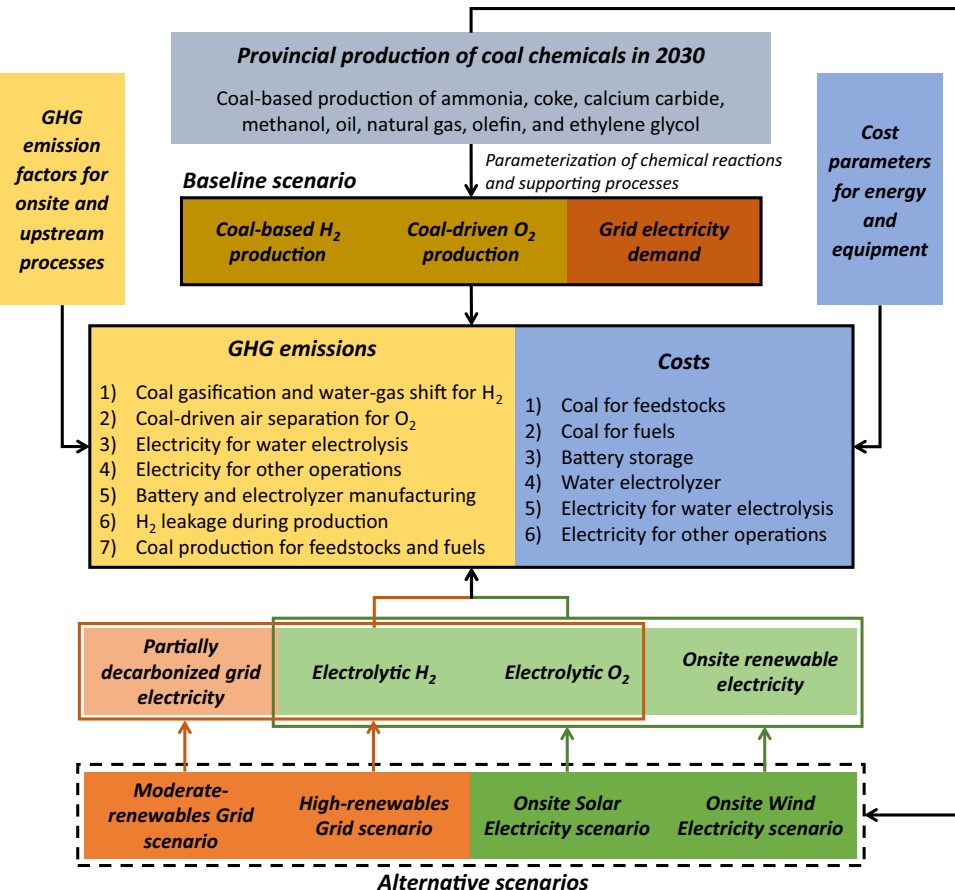

**Fig. 6 | Estimation of greenhouse gas emissions and costs in the baseline and four alternative scenarios.** GHG greenhouse gas.

without considering operational parameter variations at a monthly or even finer resolution at the plant level. This limits the carbon and cost implications for coal chemical plant operations in practice.

All modeling procedures with data sources are provided in the Supplementary Information. We provide detailed equations and parameters in the Supplementary Tables and explain how to derive each parameter/result in the notations to each table. We arrange the Supplementary Tables in the same order as the results, namely GHG emission accounting, GHG mitigation modeling, and cost-benefit analyses.

In detail, for each coal chemical, we estimate demands for coal-based/electrolytic $H_2$, air separation-based/electrolytic $O_2$, and grid electricity based on chemical reaction equations and technical parameters of coal gasification, the water-gas shift reaction, and other plant operations (Supplementary Tables 12–13). We use 2030 grid electricity (in the MG and HG scenarios) or onsite renewable electricity (in the SE and WE scenarios) to produce electrolytic $H_2$ and $O_2$ for replacement of coal-based $H_2$ and air separation-based $O_2$ (Supplementary Fig. 1). We also use onsite solar and wind electricity to replace purchased grid electricity by coal chemical plants in the SE and WE scenarios, respectively. We detail calculations for GHG mitigation in four alternative scenarios in Supplementary Table 14. We collect technical parameters from literature and reports, including electricity consumption factors (for water electrolysis, air separation, and coal gasification), $H_2$ leakage rates during production, and requirements for water electrolyzers and battery storage (Supplementary Table 15). Provincial renewable energy availability is considered using provincial capacity factors to derive annual and lifetime electricity generation of solar and wind power facilities. Provincial life-cycle GHG emission factors of renewable electricity generation (gCO2eq/kWh) are derived

by dividing the GHG emissions of constructing and operating the facility by its lifetime electricity generation (detailed in Supplementary Table 16).

For the four alternative scenarios, we estimate reductions in GHG emissions from the water-gas shift reaction, air separation, coal gasification, and coal production, as well as additional GHG emissions from electricity use for water electrolysis, $H_2$ leakage to the atmosphere, and manufacturing of water electrolyzers and battery storage. See parameterizations and calculations in Supplementary Tables 12–17. We compare GHG emission factors of onsite chemical reactions, onsite fuel combustion, and upstream processes for each coal chemical in the SE and WE scenarios to the baseline scenario, as shown in Supplementary Table 18. We also quantify the remaining GHG emissions from coal chemical production in the SE and WE scenarios (Supplementary Table 19).

**Cost benefit analysis**

We identify cost changes in the four alternative scenarios relative to the baseline scenario to analyze the cost-effectiveness of deploying grid or renewables-based water electrolysis for China's coal chemical production. We estimate the capital and operating costs of water electrolyzers, renewable electricity facilities, and battery storage required for producing each ton of electrolytic $H_2$ (CNY/t$H_2$) as described in Supplementary Tables 20–21. We use these cost parameters with electrolytic $H_2$ demand for each coal chemical to derive the cost additions for water electrolyzers, renewable electricity generation, and battery storage (Supplementary Table 22). We quantify the cost reductions in coal use as both a feedstock and fuel in four alternative scenarios due to reductions in coal-based $H_2$ production as well as coal-driven air separation and coal gasification. We also

estimate the cost reductions resulting from using renewable electricity to replace purchased grid electricity for other plant operations in the SE and WE scenarios. Prices of coal for feedstocks, coal for fuels, and grid electricity are in Supplementary Table 20. Therefore, we derive net cost changes of four alternative scenarios relative to the baseline scenario (Table 2 and Supplementary Table 22). We also provide low and high estimates of cost changes due to the uncertainty of energy and equipment prices as well as various discount rates used for accounting (Table 2 and Supplementary Tables 20–21).

We further target provincial options (to use either solar or wind-based water electrolysis) to derive the solutions for the maximum national GHG mitigation and the minimum national cost (Supplementary Table 23). We use the Onsite Solar Electricity and Onsite Wind Electricity scenarios to optimize the combinations of provincial options based on the results of provincial GHG mitigation and costs. We then assign each provincial region to either solar or wind-based water electrolysis based on which technology yields larger GHG mitigation or lower costs.

### Reporting summary
Further information on research design is available in the Nature Portfolio Reporting Summary linked to this article.

## Data availability
The source data that support all figures in the main text and Supplementary Information are provided as a Source Data file. Source data are provided with this paper.

## Code availability
The modeling procedures with detailed parameterizations and formulations are provided in the Supplementary Information.

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

## Acknowledgements

Y.G., L.P., and D.L.M. acknowledge support from the Ma Huateng Foundation's grant to Princeton University and the Princeton University Library Open Access Fund. Y.G. acknowledges support from the Schmidt Science Fellows, in partnership with the Rhodes Trust. J.T. acknowledges support from the Ministry of Ecology and Environment of China through project 2022-YRUC-01-0403 and the National Science Foundation of China through project 72348001.

## Author contributions

Y.G. and D.L.M. conceived the research and designed the study; Y.G. compiled the dataset on China's coal chemical sector; Y.G. designed the scenarios, parameterized the life-cycle processes for coal chemical production and hydrogen production, and conducted modeling for GHG emissions, mitigation potential, and costs; L.P. and J.T. contributed to the parameterizations; Y.G., L.P., J.T., and D.L.M. analyzed the results; Y.G. and D.L.M. wrote the paper with contributions from L.P. and J.T.

## Competing interests

The authors declare no competing interests.
