## [Peer review file · Nature Communications]

REVIEWER COMMENTS

Reviewer #1 (Remarks to the Author):

This paper has done an impressive work on the deploying green hydrogen to decarbonize China's coal chemical sector through exploring GHG mitigation potential and costs. The GHG mitigation potential and costs for onsite deployment of green hydrogen, green oxygen, and renewable electricity in China's coal chemical sector are examined via deploying onsite solar and wind power combined with water electrolysis. The paper is well organized and the content is interesting. The authors provided detailed source data in supplementary information to support the data credibility, data processing and the accuracy of analysis and conclusion. I believe this paper has a certain guiding significance for the development of China's coal chemical industry in the 14th Five-Year period. I recommend this paper to be published in Nature Communications after a revision.

Major:

However, there is a key issue which needs the authors to think and practice. This issue is how to realize the coupling and integration of coal chemical sector and renewable energy. The renewable energy, e.g., wind and solar energy, is intermittent and volatile, while the coal chemical sector is continuous and stable. How to achieve optimal matching between the renewable energy and coal chemical sector. This is a system engineering problem. Different matching will lead to large influence of the GHG mitigation potential and costs. As the amount of renewable wind and solar energy in different provinces in China varies greatly, and the development of the coal chemical industry is also different, i.e., resources, policies, etc. The models of GHG mitigation potential and costs used in this paper are linear models, which cannot really reflect the volatility of both the supply side and the demand side. In fact, I hope the authors can finally give some constructive suggestions and prospects, such as how to integrate coal chemical industry and renewable energy in different provinces or regions (composed of several provinces) in China, and the degree of renewable energy alternative in the medium and long-term development of coal chemical industry.

Minor:

The methods section in this paper should be put before the conclusion.

It is not clear how the GHG mitigation potential and costs are calculated. Please provide the detailed calculation models of the GHG mitigation potential and costs in the Supplementary Information.

I am not sure if CNY is acceptable by the journal.

Format the reference in Line 501, Page 21. Provide the website source.

Reviewer #2 (Remarks to the Author):

The authors estimate China's coal chemical production resulted in substantial GHG emissions and discuss the effect of whether to use onsite solar or wind power for water electrolysis on GHG emissions and coal chemical product costs. The onsite deployment of renewables-based electrolytic H₂ and O₂ to replace coal-based H₂ and air separation-based O₂ in the coal chemical sector in the onsite solar electricity (SE) scenario and onsite wind electricity (WE) scenario can reduce 50% and 52%, respectively, of 2030 baseline GHG emissions from coal chemical production in China. The results are very instructive for the decarbonization pathway for China's coal chemical sector. In my opinion, the manuscript is adapted for publication upon some minor modifications, as detailed in the following.

1. In Page 6, the authors find that "MG and HG scenarios increase GHG emissions by 35% and 15% (448 and 183 MtCO₂eq), respectively, relative to the baseline scenario". "However, the SE and WE scenarios substantially reduce GHG emissions by 50% and 52% (632 and 659 MtCO₂eq), respectively, relative to the baseline scenario". This result may be related to CO₂ emissions from grid electricity consists of coal, hydro, wind, solar nuclear, gas and biomass and others. Please provide a supplementary explanation of the CO₂ emission factors of various types of electricity.

2. There is no production of traditional coal chemicals and modern coal chemicals in Beijing or Tibet. So there seems to be no need for water electrolysis for the coal chemical sector in these two regions.

This is also consistent with Supplementary Table S14. However, in Page 8, the authors inference that most provinces "increase their GHG emissions relative to the baseline scenario" and the Figure 3 show the increase of CO2 emission for Beijing and Tibet. The inconsistency of the data is confusing.

3. In Page 9, "However, Jiangsu, Zhejiang, and Jiangxi slightly increase their GHG emissions",, "This is because the three provinces produce the majority (~60%) of solar panels in China which results in additional GHG emissions". Whether are there duplicate statistics for CO2 emissions of solar panels and the life-cycle GHG emission of solar/wind electricity?

In particular, explain the source and composition of the data in Figure 3.

4. The meaning of "Electricity for other operations" in Figure 2 and Table 2 is unclear, how is the data obtained? Please provide a supplementary explanation.

5. Solar radiation and wind speed vary from province to province in China. The authors suggest that "provinces determine whether to use onsite solar or wind power for water electrolysis based on their lowest cost options". How do the authors take into account the natural resource condition?

6. Electricity price is an important basis to discuss the cost of hydrogen production by water electrolysis. In Page 14, "Since onsite renewable electricity is cheaper than grid electricity, GHG mitigation costs when using renewables-based water electrolysis are much lower than when using grid electricity-based". However, according to Supplementary Table S20, "an operating cost of 33 USD/kW in 2030", the price of electricity is much higher than that of grid electricity. The cost parameters for provincial solar and wind electricity generation are much lower than that of grid electricity as shown in Supplementary Table S21. Data consistency in calculating costs needs to be explained.

7. Please state what "+" "-" represent in the Table 2.

Reviewer #3 (Remarks to the Author):

This paper gives detailed analysis of GHG emission and cost about deploying green hydrogen to decarbonize China's coal chemical sector.

Two comments:

1. What is the scientific value of this study? Please give more clear and detailed conclusions.
2. In the analysis, hydrogen storage and transportation seems not be considered. Will they affect the carbon emission and cost of green hydrogen to decarbonize coal chemical sector?

Reviewer #5 (Remarks to the Author):

The present paper investigates examine the GHG mitigation potential and costs for onsite deployment of green hydrogen, green oxygen, and renewable electricity in China's coal chemical sector, via deploying onsite solar and wind power combined with water electrolysis. The topic presented in this work is really interesting. However, several modifications are required:

I analyze the single sections:

Abstract has inappropriate structure. I suggest to answer the following aspects: - general context - novelty of the work - methodology used (describe briefly the main methods or treatments applied) - main results and related interpretations.

Introduction: This section should briefly place the study in a wide context and emphasize why it is relevant carrying out the analysis. It should define the purpose of the work and its significance. In this perspective, this section is too succinct and fails to effectively point out the relevance of your contribution towards the existing literature. The authors could better introduce the topic by framing it within the energy/sustainability transition dynamics. Some literature is also required to justify the focus on green hydrogen, among the others. Please see:

<https://doi.org/10.1016/j.cogsc.2021.100506>

<https://doi.org/10.1016/j.respol.2021.104464>
<https://link.springer.com/article/10.1007/s40888-020-00206-4>

<https://doi.org/10.1016/j.eneco.2023.106642>

Materials and methods: I found this section very important for the readability of the paper. I think the research procedure could be much more clearly described by means of a diagram also highlighting its potential and limit.

Discussions: The discussion of the results is merely descriptive and the obtained evidence is flimsy due to the fact the outcomes are not supported by an adequate discussion in light of scientific literature. Authors should discuss the results and how they can be interpreted in perspective of previous studies and their implications should be discussed in the broadest context possible.

Conclusions: Conclusions must also be revised according to the previous comments. In particular, they should discuss practical and policy implications as well as future lines of research. As it stands now, they fail to extract all the juice of your work.

I hope these comments might help in improving the paper and encourage the authors to move forward.

Our responses (in black) together with *the reviewers' comments (in black and italic)* and **changes to the paper (in blue)** are presented below. Revisions to the manuscript are either highlighted or in tracked changes. For ease of identifying the changes we made, we have submitted versions with and without tracked modifications of the revised manuscript. **Line and Page numbers** (in black and bold) below refer to the manuscript version with tracked modifications.

Responses to Reviewer #1

Comment 1:

This paper has done an impressive work on the deploying green hydrogen to decarbonize China's coal chemical sector through exploring GHG mitigation potential and costs. The GHG mitigation potential and costs for onsite deployment of green hydrogen, green oxygen, and renewable electricity in China's coal chemical sector are examined via deploying onsite solar and wind power combined with water electrolysis. The paper is welly organized and the content is interesting. The authors provided detailed source data in supplementary information to support the data credibility, data processing and the accuracy of analysis and conclusion. I believe this paper has a certain guiding significance for the development of China's coal chemical industry in the 14th Five-Year period. I recommend this paper to be published in Nature Communications after a revision.

Reply:

We appreciate your enthusiasm about our work. We carefully studied your comments and revised the paper accordingly below.

Comment 2:

However, there is a key issue which needs the authors to think and practice. This issue is how to realize the coupling and integration of coal chemical sector and renewable energy. The renewable energy, e.g., wind and solar energy, is intermittent and volatile, while the coal chemical sector is

continuous and stable. How to achieve optimal matching between the renewable energy and coal chemical sector. This is a system engineering problem. Different matching will lead to large influence of the GHG mitigation potential and costs.

As the amount of renewable wind and solar energy in different provinces in China varies greatly, and the development of the coal chemical industry is also different, i.e., resources, policies, etc. The models of GHG mitigation potential and costs used in this paper are linear models, which cannot really reflect the volatility of both the supply side and the demand side.

In fact, I hope the authors can finally give some constructive suggestions and prospects, such as how to integrate coal chemical industry and renewable energy in different provinces or regions (composed of several provinces) in China, and the degree of renewable energy alternative in the medium and long-term development of coal chemical industry.

Reply:

Thanks for this insightful comment regarding how to present the integration of variable renewable resources and coal chemical industry among China's provinces and your request to provide more direct policy implications.

To address your suggestions, we first explain in more detail the solar and wind availability across provinces in China in the Results, **Lines 173-183, Page 8**: Our model includes the availability of solar and wind resources in each province. It uses provincial solar/wind capacity factors to derive provincial GHG mitigation and provincial costs of deploying onsite renewable electricity generation, green H₂, and green O₂ in the coal chemical sector. We normalize GHG emissions and costs of manufacture, installation, and operation of solar/wind power facilities in the SE/WE (Onsite Solar Electricity / Onsite Wind Electricity) scenario over the lifetime electricity generation of each facility to obtain results per kWh generation. We consider provincial renewable energy availability by including provincial capacity factors (= annual electricity generation ÷ rated maximum electricity generation) of solar/wind power facilities in calculating their lifetime electricity generation. We thus derive GHG emissions and costs per kWh of solar/wind electricity generated for each province across China (see Supplementary Tables 16 and 21).

We also describe this approach in more detail in the Methods, **Lines 580-585, Page 25**: Provincial renewable energy availability is considered using provincial capacity factors to derive annual and lifetime electricity generation of solar and wind power facilities. Provincial life-cycle GHG

emission factors of renewable electricity generation (gCO₂eq/kWh) are derived by dividing the GHG emissions of constructing and operating the facility by its lifetime electricity generation (detailed in Supplementary Table 16).

Second, we clarify the inclusion of battery storage for renewable electricity in the Results section, **Lines 183-187, Page 8**: As coal chemical production requires continuous H₂ supply, our model includes battery storage to address the intermittency of solar and wind energy. Battery storage is used to provide stable electricity from renewables to continuously generate H₂ and O₂ via water electrolysis¹ for chemical syntheses. We quantify the GHG emissions and costs of battery storage for renewable electricity in the SE/WE scenario (detailed in Figure 2, Table 2, Supplementary Tables 15 and 20).

Third, we propose provincial options to integrate either solar or wind-based H₂ into each provincial coal chemical sector in the Discussion section, **Lines 394-420, Pages 18-19**: We explore the best options for each province to decarbonize the coal chemical sector via either solar or wind-based H₂ to achieve the maximum national GHG mitigation and minimum national costs. We use results from the Onsite Solar Electricity and Onsite Wind Electricity scenarios to optimize the combinations of provincial options to maximize national GHG mitigation and to minimize national costs. We assign each province to either solar or wind-based water electrolysis based on which technology yields larger GHG mitigation or lower costs within each province. In the maximum national GHG mitigation solution, we find a reduction of -57% (-722 MtCO₂eq) relative to 2030 baseline GHG emissions at a national annual cost addition of 26 billion CNY (36 CNY/tCO₂eq) relative to baseline costs. In this transition, 17 provinces deploy solar- and 12 provinces deploy wind-based water electrolysis (excluding Beijing and Tibet where no coal chemicals are produced, and Hong Kong, Macau, and Taiwan where no coal chemical data are available). In the minimum national cost solution, we find a national annual cost increase of 6.3 billion CNY (9.4 CNY/tCO₂eq) in 2030 relative to baseline costs with national GHG mitigation of -53% (-665 MtCO₂eq) relative to baseline emissions. In this transition, 26 provinces deploy solar- and 3 provinces deploy wind-based water electrolysis, with the same exclusions as above. These two solutions have similar GHG mitigation (-57% vs. -53% relative to the baseline) but costs for the minimum cost solution are only 24% of those for the maximum mitigation solution. We thus suggest that provinces determine whether to use onsite solar or wind power for water electrolysis based on their lowest cost options (see Supplementary Table 23). We further find that Inner Mongolia, Shaanxi, Ningxia,

and Xinjiang have much larger GHG mitigation potential than other provinces while simultaneously achieving net cost reductions due to their abundant solar energy. These four provinces collectively account for 52% of total GHG mitigation in the minimum national cost solution. For policymaking and demonstration projects, the four provinces can be pioneers in cost-effectively deploying onsite green H₂ and O₂ in coal chemical production. This clearly shows the enormous potential of decarbonizing the coal chemical sector at only a small cost increase at the national level.

Fourth, we discuss the prospects for future renewable electricity use in the coal chemical sector in the Discussion section, **Lines 466-478, Pages 20-21**: China's coal chemical plants generally have onsite captive coal power plants to generate heat and electricity for chemical production, with grid electricity as a supplementary power source². In this study, we use onsite renewable electricity to electrolyze the water and to replace grid electricity purchased by coal chemical plants. The onsite coal power plants remain operational as a high-temperature heat source and as an electricity source for plant operations besides water electrolysis. China's chemical sector is expected to be included in the national carbon trading market by 2035³, and high-temperature heat generation from coal is hard to replace with renewable electricity at scale in the near future. Therefore, we propose the onsite deployment of renewable electricity for water electrolysis and for replacement of grid electricity use in coal chemical plants during 2023-2035 but retain the onsite coal power plants for other plant operations. As electrification technologies advance over the next decade, we suggest that onsite deployment of renewable electricity should increasingly replace onsite heat and power generation from coal for industrial processes (such as air separation and coal gasification).

References:

1. Palmer, G., Roberts, A., Hoadley, A., Dargaville, R., Honnery, D. Life-cycle greenhouse gas emissions and net energy assessment of large-scale hydrogen production via electrolysis and solar PV. *Energy & Environmental Science* **14**, 5113-5131 (2021).
2. Zhang, Y. *et al.* Intensive carbon dioxide emission of coal chemical industry in China. *Applied Energy* **236**, 540-550 (2019).
3. Goulder, L. H., Long, X., Qu, C., Zhang, D. China's Nationwide CO₂ Emissions Trading System: A General Equilibrium Assessment. (Global Trade Analysis Project Resources, 2023); <https://gtap.agecon.purdue.edu/resources/download/11676.pdf>

Comment 3:

The methods section in this paper should be put before the conclusion.

Reply:

The journal format requires the Methods section to come after the Discussion. Accordingly, we need to place the Methods at the end of the main text.

Comment 4:

It is not clear how the GHG mitigation potential and costs are calculated. Please provide the detailed calculation models of the GHG mitigation potential and costs in the Supplementary Information.

Reply:

We have added additional information on how we calculate GHG mitigation and cost changes including a figure added to the Methods section and additional details in the Supplementary Information. Since modeling of GHG mitigation and costs includes intensive parameterizations and formulations, we include only brief descriptions in the Methods section for GHG emission accounting in 2020 and the 2030 baseline scenario, GHG mitigation modeling in four alternative scenarios, and cost-benefit analyses for the four alternative scenarios. We have added clarifications on the calculation procedures below to the Supplementary Information as **Supplementary Note: Methods**. We also have added a diagram to clarify our modeling as in **Figure 6**.

Figure 6. Estimation of GHG emissions and costs in the baseline and four alternative scenarios.

Supplementary Note: Methods

Supplementary Tables and related calculations are presented in the order of the Results in the main text: 1) GHG emission accounting, 2) GHG mitigation modeling, and 3) Cost-benefit analyses. We provide our modeling framework in Figure 6.

GHG Emissions Accounting: We track GHG emissions from coal chemical production using 2020 chemical production quantities (*Supplementary Tables 1-3*) and 2030 projected chemical production quantities (*Supplementary Tables 4-6*) coupled with GHG emission factors of coal chemical production in 2020 (*Supplementary Tables 7-10*) and in the 2030 baseline scenario (*Supplementary Table 11*), respectively.

Second, we model the GHG mitigation potential of four alternative scenarios based on parametrizations of chemical reactions and supporting industrial processes, detailed in *Supplementary Tables 12-19*. Specifically, we use projected 2030 provincial production of coal chemicals (*Supplementary Table 6*) and demand per ton of coal chemicals for electrolytic H₂ and

O₂ (*Supplementary Table 13*) and for grid electricity (*Supplementary Table 8*) to estimate the provincial demand for electrolytic H₂ and O₂ and grid electricity (that can be replaced by onsite renewable electricity). Then we characterize GHG emission changes by province and by individual process using these provincial demands with GHG emission factors of all major chemical reactions required the production of the eight coal chemicals analyzed (*Supplementary Tables 13*), coal production, 2030 grid electricity generation (*Supplementary Table 9*), air separation, coal gasification, electrolyzer and battery manufacturing, H₂ leakage (*Supplementary Table 15*), and provincial solar/wind electricity generation (*Supplementary Table 16*). We present provincial GHG emission changes of four alternative scenarios relative to the baseline scenario by individual process in *Supplementary Table 14* and we detail how to attribute GHG emission changes of individual processes to provinces in *Supplementary Tables 14 and 17*. We present national average GHG emission factors of coal chemical production in the SE/WE scenarios in *Supplementary Table 18* and remaining GHG emissions in the SE/WE scenarios in *Supplementary Table 19*.

Third, we collect or estimate the cost parameters for coal chemical production and water electrolysis in *Supplementary Table 20*. We estimate the cost parameters for provincial solar/wind electricity generation in *Supplementary Table 21*. We then use these cost parameters with the provincial demands for electrolytic H₂ and O₂ and additional grid/onsite renewable electricity derived above to estimate provincial cost changes by individual processes in the SE/WE scenarios (*Supplementary Table 22*).

In addition, we derive the provincial options for either solar or wind-based water electrolysis to achieve the maximum GHG mitigation or minimum cost of national coal chemical production (*Supplementary Table 23*) to provide provincial-level policy implications. We also analyze the GHG mitigation and economic benefits of selling excess green O₂ (*Supplementary Table 24*) and the land area needed for renewables in the SE/WE scenarios (*Supplementary Table 25*).

In the Methods section of the main text, **Lines 563-567, Page 25**, we add: All modeling procedures with data sources are provided in the Supplementary Information. We provide detailed equations and parameters in the Supplementary Tables and explain how to derive each parameter/result in the notations to each table. We arrange the Supplementary Tables in the same order as the results, namely GHG emission accounting, GHG mitigation modeling, and cost-benefit analyses.

Comment 5:

I am not sure if CNY is acceptable by the journal.

Reply:

We have checked out published papers in the journal and found the metric of CNY is acceptable for presentation of results.

Comment 6:

Format the reference in Line 501, Page 21. Provide the website source.

Reply:

Thanks for the note. We have added the website to this reference.

Responses to Reviewer #2

Comment 1:

The authors estimate China's coal chemical production resulted in substantial GHG emissions and discuss the effect of whether to use onsite solar or wind power for water electrolysis on GHG emissions and coal chemical product costs. The onsite deployment of renewables-based electrolytic H₂ and O₂ to replace coal-based H₂ and air separation-based O₂ in the coal chemical sector in the onsite solar electricity (SE) scenario and onsite wind electricity (WE) scenario can reduce 50% and 52%, respectively, of 2030 baseline GHG emissions from coal chemical production in China. The results are very instructive for the decarbonization pathway for China's coal chemical sector. In my opinion, the manuscript is adapted for publication upon some minor modifications, as detailed in the following.

Reply:

Thanks for your endorsement of our work. We appreciate your valuable comments and have revised the paper accordingly as follows.

Comment 2:

In Page 6, the authors find that “MG and HG scenarios increase GHG emissions by 35% and 15% (448 and 183 MtCO₂eq), respectively, relative to the baseline scenario”. “However, the SE and WE scenarios substantially reduce GHG emissions by 50% and 52% (632 and 659 MtCO₂eq), respectively, relative to the baseline scenario”. This result may be related to CO₂ emissions from grid electricity consists of coal, hydro, wind, solar, nuclear, gas and biomass, and others. Please provide a supplementary explanation of the CO₂ emission factors of various types of electricity.

Reply:

We clarify the 2020/2030 grid electricity mixes and their GHG emission factors in the Results, **Lines 154-170, Pages 7-8**: The 2020 grid electricity is generated by: coal (61%), hydro (18%), wind (6%), solar (3%), nuclear (5%), gas (3%), and biomass and others (4%)¹⁶, with a life-cycle GHG emission factor of 586 kg CO₂eq/MWh. The 2030 Moderate-renewables power grid (572 kg CO₂eq/MWh) decreases the coal share to 57% and increases the wind and solar shares to 7% and

5%, respectively, with other energy sources making up the rest of generation (hydro, 14%; nuclear, 8%; gas, 7%; biomass and others, 2%), projected by the International Energy Agency¹⁷. The 2030 High-renewables power grid (441 kg CO₂eq/MWh) is further decarbonized relative to the 2030 Moderate-renewables power grid, resulting in contributions to generation as follows: coal (43%), wind (16%), solar (9%), hydro (14%), nuclear (7%), gas (6%), and others (5%), projected by an integrated model for the power sector conducted by China’s state-owned power companies¹⁸. The Moderate-renewables Grid (MG) and High-renewables Grid (HG) scenarios use the Moderate- and High-renewables grid electricity in 2030 for water electrolysis, respectively, to produce electrolytic H₂ and O₂ for coal chemical production.

We provide the life-cycle GHG emission factors of various electricity generation technologies in Supplementary Table 9. We present the 2030 electricity mix projections in Supplementary Table 10.

Supplementary Table 9 GHG emission factors for fuels, electricity, and heat

GHG emission factor		Value	Metric	Source
Onsite	Coal combustion	1.99	kgCO ₂ eq/kg	Ref ²³
	Natural gas combustion	2.16	kgCO ₂ eq/m ³	Ref ²³
Life-cycle	Coal production for fuel	0.180	kgCO ₂ eq/kg	Ref ¹⁴
	Coal production for feedstock	0.288	kgCO ₂ eq/kg	Ref ¹⁴
	Natural gas production for fuel	0.279	kgCO ₂ eq/m ³	Ref ¹⁴
	Outsourced heat supply	0.126	kgCO ₂ eq/MJ	Ref ¹⁴
	Solar electricity	0.0362	tCO ₂ eq/MWh	Ref ²⁶
	Wind electricity (onshore)	0.0199	tCO ₂ eq/MWh	Ref ²⁷
	2020 grid electricity	0.586	tCO ₂ eq/MWh	*
	2030 moderate-renewables grid electricity	0.572	tCO ₂ eq/MWh	*
	2030 high-renewables grid electricity	0.441	tCO ₂ eq/MWh	*

*Note: We estimate the life-cycle GHG emission factors of 2020/2030 grid electricity using grid electricity mixes (Supplementary Table 10). We use a national average thermal power generation efficiency of 40.2%²⁸ for fuel-to-electricity. Thus, life-cycle GHG emission factor of grid electricity = Share of coal-based electricity ÷ 40.2% × (onsite GHG emission factor of coal combustion + life-cycle GHG emission factor of coal production for fuel) + share of natural gas-based electricity ÷ 40.2% × (onsite GHG emission factor of natural gas combustion + life-cycle GHG emission factor of natural gas production for fuel) + share of solar electricity × life-cycle GHG emission factor of solar electricity + share of wind electricity × life-cycle GHG emission factor of wind electricity.

Supplementary Table 10 Grid electricity mixes for 2020 and 2030

	Coal	Hydro	Nuclear	Wind	Solar	Gas	Other
2020 ^{Ref 29}	0.61	0.18	0.05	0.06	0.03	0.03	0.04
2030 Moderate-renewables ^{Ref 30}	0.57	0.14	0.08	0.07	0.05	0.07	0.02
2030 High-renewables ^{Ref 31}	0.43	0.14	0.07	0.16	0.09	0.06	0.05

Note: We use the 2020 grid electricity mix for estimation of 2020 coal chemical GHG emissions. We use the 2030 Moderate-renewables grid electricity mix for estimation of GHG emissions in the 2030 baseline scenario and 2030 Moderate-renewables Grid Electricity scenario, and use the 2030 High-renewables grid electricity mix for estimation of GHG emissions in the 2030 High-renewables Grid Electricity scenario. Total electricity generation in China was ~7600 TWh in 2020²⁹ and is projected to be 9400-11800 TWh in 2030^{30,31}.

Comment 3:

There is no production of traditional coal chemicals and modern coal chemicals in Beijing or Tibet. So there seems to be no need for water electrolysis for the coal chemical sector in these two regions. This is also consistent with Supplementary Table S14. However, in Page 8, the authors inference that most provinces “increase their GHG emissions relative to the baseline scenario” and Figure 3 shows the increase of CO₂ emission for Beijing and Tibet. The inconsistency of the data is confusing.

Reply:

Thanks for noting the need for this correction. Yes, there is no coal chemical production in Beijing and Tibet. However, Beijing and Tibet do have GHG emission changes in the four alternative scenarios relative to the baseline, due to grid electricity generation and/or solar panel production for other regions. We now clarify this as in **Lines 254-259, Page 11: In the MG and HG scenarios, most provinces increase their GHG emissions relative to the baseline scenario because their electricity generation increases as grid electricity demand increases for water electrolysis. Beijing and Tibet, although without coal chemical production, provide a portion of grid electricity for grid-based water electrolysis occurring in other provinces in the MG and HG scenarios, which increases their GHG emissions relative to the baseline scenario.**

and in **Lines 272-275, Page 12: Beijing also provides solar panels for solar-based water electrolysis occurring in other provinces in the SE scenario, but its avoided emissions from**

reductions in grid electricity generation more than offset those from its solar panel production. Therefore, Beijing decreases GHG emissions in the SE scenario relative to the baseline scenario.

We clarify the attribution method for GHG emission changes in the Results, **Lines 242-253, Page 11**: We attribute GHG emission changes in four alternative scenarios to provinces where variations in industrial processes physically occur (Figure 3). Changes in onsite GHG emissions occur in provinces where coal chemicals are produced, including those from chemical processes, onsite fuel combustion, and H₂ leakage. We allocate GHG emission changes in coal production, grid electricity generation, and solar power facility manufacturing to provinces based on provincial production of coal, thermal power, and solar panels, respectively (see Supplementary Table 17)^{1,26}. We attribute GHG emissions from wind power facility manufacturing to provinces where wind turbines are deployed, since building materials and nacelles (mainly consisting of concrete and steel) account for major GHG emissions²⁷ and they have short cost-effective transport distances. We also assume that electrolyzer and battery manufacturing occurs in local provinces. Such an attribution approach for upstream environmental impacts has been applied in related studies^{28,29}. We detail the attribution method in Supplementary Table 14.

Therefore, Figure 3 presents the aggregate results of GHG emission changes in all individual processes for each province. The original version of Figure 3 with its source data is correct.

However, we revised Supplementary Table 14. This table originally used the results without allocating upstream GHG emission changes in grid electricity generation, solar panel manufacturing, and coal production into provinces where these changes physically occur. We have corrected Supplementary Table 14 using the results that allocate upstream GHG emission changes into provinces (see highlighted columns in Supplementary Table 14).

Comment 4:

In Page 9, “However, Jiangsu, Zhejiang, and Jiangxi slightly increase their GHG emissions”, ..., “This is because the three provinces produce the majority (~60%) of solar panels in China which results in additional GHG emissions”. Whether are there duplicate statistics for CO₂ emissions of solar panels and the life-cycle GHG emission of solar/wind electricity? In particular, explain the source and composition of the data in Figure 3.

Reply:

We account for the life-cycle GHG emissions of solar/wind electricity only once in our model. We use the quantity of solar panels produced in each province to allocate national life-cycle GHG emissions of solar electricity generation into provinces. We clarify this attribution method for GHG emissions **in the response to Comment 3 as above**.

The detailed data for Figure 3 are provided in the Source Data file. The source data for Figure 3 are the aggregate results of Supplementary Table 14. We detail the modeling procedures in the notations to Supplementary Table 14. We also clarify the overall modeling framework of GHG emissions and costs of various scenarios in the Methods, Figure 6, and Supplementary Note: Methods.

Comment 5:

The meaning of “Electricity for other operations” in Figure 2 and Table 2 is unclear, how is the data obtained? Please provide a supplementary explanation.

Reply:

Thanks. We clarify “*Electricity for other operations*” in Figure 2 (**Lines 239-241, Page 11**) and Table 2 (**Lines 332-334, Page 15**): *Electricity for other operations* refers to electricity consumption in coal chemical plants used for any process except coal gasification, air separation, or water electrolysis (such as for the water-gas shift reaction).

We clarify the calculations for GHG emission changes and cost changes in *electricity generation for other operations* in Supplementary Table 14 and Supplementary Tables 20-21, respectively.

Comment 6:

Solar radiation and wind speed vary from province to province in China. The authors suggest that “provinces determine whether to use onsite solar or wind power for water electrolysis based on their lowest cost options”. How do the authors take into account the natural resource condition?

Reply:

We explain in detail the provincial solar and wind availability in the Results, **Lines 173-183, Page 8**: Our model includes the availability of solar and wind resources in each province. It uses provincial solar/wind capacity factors to derive provincial GHG mitigation and provincial costs of deploying onsite renewable electricity generation, green H₂, and green O₂ in the coal chemical sector. We normalize GHG emissions and costs of manufacture, installation, and operation of solar/wind power facilities in the SE/WE (Onsite Solar Electricity / Onsite Wind Electricity) scenario over the lifetime electricity generation of each facility to obtain results per kWh generation. We consider provincial renewable energy availability by including provincial capacity factors (= annual electricity generation ÷ rated maximum electricity generation) of solar/wind power facilities in calculating their lifetime electricity generation. We thus derive GHG emissions and costs per kWh of solar/wind electricity generated for each province across China (see Supplementary Tables 16 and 21).

We also clarify this in the Methods, **Lines 580-585, Page 25**: Provincial renewable energy availability is considered using provincial capacity factors to derive annual and lifetime electricity generation of solar and wind power facilities. Provincial life-cycle GHG emission factors of renewable electricity generation (gCO₂eq/kWh) are derived by dividing the GHG emissions of constructing and operating the facility by its lifetime electricity generation (detailed in Supplementary Table 16).

We propose provincial options to integrate either solar or wind-based H₂ into each provincial coal chemical sector in the Discussion section, **Lines 394-420, Pages 18-19**: We explore the best options for each province to decarbonize the coal chemical sector via either solar or wind-based H₂ to achieve the maximum national GHG mitigation and minimum national costs. We use results from the Onsite Solar Electricity and Onsite Wind Electricity scenarios to optimize the combinations of provincial options to maximize national GHG mitigation and to minimize national costs. We assign each province to either solar or wind-based water electrolysis based on which technology yields larger GHG mitigation or lower costs within each province. In the maximum national GHG mitigation solution, we find a reduction of -57% (-722 MtCO₂eq) relative to 2030 baseline GHG emissions at a national annual cost addition of 26 billion CNY (36 CNY/tCO₂eq) relative to baseline costs. In this transition, 17 provinces deploy solar- and 12 provinces deploy wind-based water electrolysis (excluding Beijing and Tibet where no coal chemicals are produced, and Hong Kong, Macau, and Taiwan where no coal chemical data are available). In the minimum

national cost solution, we find a national annual cost increase of 6.3 billion CNY (9.4 CNY/tCO₂eq) in 2030 relative to baseline costs with national GHG mitigation of -53% (-665 MtCO₂eq) relative to baseline emissions. In this transition, 26 provinces deploy solar- and 3 provinces deploy wind-based water electrolysis, with the same exclusions as above. These two solutions have similar GHG mitigation (-57% vs. -53% relative to the baseline) but costs for the minimum cost solution are only 24% of those for the maximum mitigation solution. We thus suggest that provinces determine whether to use onsite solar or wind power for water electrolysis based on their lowest cost options (see Supplementary Table 23). We further find that Inner Mongolia, Shaanxi, Ningxia, and Xinjiang have much larger GHG mitigation potential than other provinces while simultaneously achieving net cost reductions due to their abundant solar energy. These four provinces collectively account for 52% of total GHG mitigation in the minimum national cost solution. For policymaking and demonstration projects, the four provinces can be pioneers in cost-effectively deploying onsite green H₂ and O₂ in coal chemical production. This clearly shows the enormous potential of decarbonizing the coal chemical sector at only a small cost increase at the national level.

Comment 7:

Electricity price is an important basis to discuss the cost of hydrogen production by water electrolysis. In Page 14, “Since onsite renewable electricity is cheaper than grid electricity, GHG mitigation costs when using renewables-based water electrolysis are much lower than when using grid electricity-based”. However, according to Supplementary Table S20, “an operating cost of 33 USD/kW in 2030”, the price of electricity is much higher than that of grid electricity. The cost parameters for provincial solar and wind electricity generation are much lower than that of grid electricity as shown in Supplementary Table S21. Data consistency in calculating costs needs to be explained.

Reply:

Thanks for the note. “33 USD/kW” refers to operating costs (excluding electricity costs for water electrolysis) per kW (rather than kWh) of water electrolyzers. The costs per MWh of solar or wind electricity are derived in Supplementary Table 21 by dividing capital and operating costs of solar or wind facilities with lifetime electricity generation. The results in Supplementary Table 21

indicate lower prices of solar/wind electricity than grid electricity in all provinces. We double checked and confirm that the data consistency is correct.

We also add clarifications in **Lines 377-381, Page 17**: Since onsite renewable electricity (national average of 109 CNY/MWh for solar and 155 CNY/MWh for wind, see Supplementary Table 21) in 2030 is cheaper than grid electricity (580 CNY/MWh, see Supplementary Table 20), GHG mitigation costs when using renewables-based water electrolysis are much lower than when using grid electricity-based.

Comment 8:

Please state what “+” “-” represent in the Table 2.

Reply:

We state it in the notations to Table 2, **Line 328, Page 15**: “+/-” refers to increases/decreases in costs.

Responses to Reviewer #3

Comment 1:

This paper gives detailed analysis of GHG emission and cost about deploying green hydrogen to decarbonize China's coal chemical sector.

Reply:

We appreciate your valuable suggestions to improve the manuscript and have revised it accordingly as follows.

Comment 2:

What is the scientific value of this study? Please give more clear and detailed conclusions.

Reply:

We revised the end of the Introduction to more clearly state the scientific value of our study, in **Lines 90-99, Pages 4-5**: Here we explore the GHG mitigation potential and costs to decarbonize China's coal chemical sector through the onsite use of renewable electricity to produce decarbonized H₂ and O₂ and displace carbon-intensive grid-electricity. Onsite use of green H₂ in the coal chemical sector is a win-win opportunity. First, green H₂ can be used in the coal chemical sector for carbon-free feedstocks. Second, the coal chemical sector, which uses the most H₂ of any sector in China, will facilitate scale-up and cost reductions for green H₂ production. Our study provides specific implications for deploying renewables-based H₂, O₂, and electricity to produce a variety of coal based chemicals, and projects the lowest cost options for each province from now to about 2030. Our work goes beyond previous research to examine the role of green H₂ in decarbonizing the coal chemical sector nationally.

We revised the last few paragraphs of the Discussion to clarify our conclusions.

First, we propose provincial options to integrate either solar or wind-based H₂ into each province's coal chemical sector in the Discussion, **Lines 394-420, Pages 18-19**: We explore the best options for each province to decarbonize the coal chemical sector via either solar or wind-based H₂ to

achieve the maximum national GHG mitigation and minimum national costs. We use results from the Onsite Solar Electricity and Onsite Wind Electricity scenarios to optimize the combinations of provincial options to maximize national GHG mitigation and to minimize national costs. We assign each province to either solar or wind-based water electrolysis based on which technology yields larger GHG mitigation or lower costs within each province. In the maximum national GHG mitigation solution, we find a reduction of -57% (-722 MtCO₂eq) relative to 2030 baseline GHG emissions at a national annual cost addition of 26 billion CNY (36 CNY/tCO₂eq) relative to baseline costs. In this transition, 17 provinces deploy solar- and 12 provinces deploy wind-based water electrolysis (excluding Beijing and Tibet where no coal chemicals are produced, and Hong Kong, Macau, and Taiwan where no coal chemical data are available). In the minimum national cost solution, we find a national annual cost increase of 6.3 billion CNY (9.4 CNY/tCO₂eq) in 2030 relative to baseline costs with national GHG mitigation of -53% (-665 MtCO₂eq) relative to baseline emissions. In this transition, 26 provinces deploy solar- and 3 provinces deploy wind-based water electrolysis, with the same exclusions as above. These two solutions have similar GHG mitigation (-57% vs. -53% relative to the baseline) but costs for the minimum cost solution are only 24% of those for the maximum mitigation solution. We thus suggest that provinces determine whether to use onsite solar or wind power for water electrolysis based on their lowest cost options (see Supplementary Table 23). We further find that Inner Mongolia, Shaanxi, Ningxia, and Xinjiang have much larger GHG mitigation potential than other provinces while simultaneously achieving net cost reductions due to their abundant solar energy. These four provinces collectively account for 52% of total GHG mitigation in the minimum national cost solution. For policymaking and demonstration projects, the four provinces can be pioneers in cost-effectively deploying onsite green H₂ and O₂ in coal chemical production. This clearly shows the enormous potential of decarbonizing the coal chemical sector at only a small cost increase at the national level.

Second, we provide prospects for future renewable electricity use in the coal chemical sector in the Discussion, **Lines 466-478, Pages 20-21**: China's coal chemical plants generally have onsite captive coal power plants to generate heat and electricity for chemical production, with grid electricity as a supplementary power source¹. In this study, we use onsite renewable electricity to electrolyze the water and to replace grid electricity purchased by coal chemical plants. The onsite coal power plants remain operational as a high-temperature heat source and as an electricity source

for plant operations besides water electrolysis. China's chemical sector is expected to be included in the national carbon trading market by 2035², and high-temperature heat generation from coal is hard to replace with renewable electricity at scale in the near future. Therefore, we propose the onsite deployment of renewable electricity for water electrolysis and for replacement of grid electricity use in coal chemical plants during 2023-2035 but retain the onsite coal power plants for other plant operations. As electrification technologies advance over the next decade, we suggest that onsite deployment of renewable electricity should increasingly replace onsite heat and power generation from coal for industrial processes (such as air separation and coal gasification).

Third, we make concluding statements to provide policy implications and prospects for future study, at the end of Discussion in **Lines 479-501, Page 21**: Onsite deployment of renewables-based electrolytic H₂ and O₂ is a feasible pathway to partially decarbonize China's coal chemical sector. We suggest that provinces determine whether to use onsite solar or wind power for water electrolysis based on their lowest cost options, which collectively reduce 53% of the 2030 baseline GHG emissions from coal chemical production at the low cost in 2030 of 9.4 CNY/tCO₂eq. We find Inner Mongolia, Shaanxi, Ningxia, and Xinjiang collectively account for 52% of total GHG mitigation that is possible with net cost reductions. These four provinces, which have extensive available land, can be pioneers in deploying cost-effective onsite green H₂ and O₂ in coal chemical production. Excess green O₂ sales can substantially reduce costs of renewables-based water electrolysis for coal chemical production. GHG mitigation costs can be offset if the coal chemical sector is included in China's carbon trading market (the carbon price is ~50 CNY/tCO₂ in 2021³, which makes it highly profitable to trade carbon permits when compared to the 9.4 CNY/tCO₂eq cost of mitigation in 2030). Coupling of chemical production and green hydrogen is a win-win opportunity to both scale up the deployment of green H₂ and to utilize a low-carbon feedstock for the coal-chemical sector. We plan to use plant-level operational data to extend the study of the coal chemical sector to examine the environmental co-benefits of using on-site green H₂ for air quality improvements and freshwater conservation. We will also consider the use of onsite renewable electricity with battery storage to replace captive coal power facilities in coal chemical plants when high-temperature heat generation from electricity is feasible at scale.

References:

1. Zhang, Y. *et al.* Intensive carbon dioxide emission of coal chemical industry in China. *Applied Energy* **236**, 540-550 (2019).
2. Goulder, L. H., Long, X., Qu, C., Zhang, D. China's Nationwide CO₂ Emissions Trading System: A General Equilibrium Assessment. (Global Trade Analysis Project Resources, 2023); <https://gtap.agecon.purdue.edu/resources/download/11676.pdf>
3. The first year of China's national carbon market, reviewed (China Dialogue, 2022); <https://chinadialogue.net/en/climate/the-first-year-of-chinas-national-carbon-market-reviewed/>

Comment 3:

In the analysis, hydrogen storage and transportation seem not be considered. Will they affect the carbon emission and cost of green hydrogen to decarbonize coal chemical sector?

Reply:

Thanks for this point. We add analyses of hydrogen storage and transport to the Discussion in **Lines 433-440, Page 19**: We suggest onsite deployment of green H₂ in coal chemical plants, which can avoid costly long-distance H₂ transport¹. Considering battery storage is more widely used than H₂ storage², we include the GHG emissions and costs of using battery storage for renewable electricity instead of H₂ storage to reduce costs and H₂ leakage. Battery storage can help provide stable renewable electricity for water electrolysis to continuously deliver green H₂ for coal chemical production. In practice, coal chemical plants may need very short pipelines for H₂ transport within plants and small-scale H₂ storage as a back-up, which results in insignificant increases of GHG emissions and costs.

Accordingly, we clarify the inclusion of battery storage for renewable electricity in the Results section, **Lines 183-187, Page 8**: As coal chemical production requires continuous H₂ supply, our model includes battery storage to address the intermittency of solar and wind energy. Battery storage is used to provide stable electricity from renewables to continuously generate H₂ and O₂ via water electrolysis³ for chemical syntheses. We quantify the GHG emissions and costs of battery storage for renewable electricity in the SE/WE scenario (detailed in Figure 2, Table 2, and Supplementary Tables 15 and 20).

References:

1. Nazir, H. *et al.* Is the H₂ economy realizable in the foreseeable future? Part II: H₂ storage, transportation, and distribution. *International journal of hydrogen energy* **45**, 20693-20708 (2020).
2. Peng, L., Mauzerall, D. L., Zhong, Y. D., He, G. Heterogeneous effects of battery storage deployment strategies on decarbonization of provincial power systems in China. *Nature Communications* **14**, 4858 (2023).
3. Palmer, G., Roberts, A., Hoadley, A., Dargaville, R., Honnery, D. Life-cycle greenhouse gas emissions and net energy assessment of large-scale hydrogen production via electrolysis and solar PV. *Energy & Environmental Science* **14**, 5113-5131 (2021).

Responses to Reviewer #5

Comment 1:

The present paper investigates the GHG mitigation potential and costs for onsite deployment of green hydrogen, green oxygen, and renewable electricity in China's coal chemical sector, via deploying onsite solar and wind power combined with water electrolysis. The topic presented in this work is really interesting. However, several modifications are required.

Reply:

We appreciate your detailed comments which helped improve our paper substantially. Our paper has been revised as follows.

Comment 2:

I analyze the single sections:

Abstract has inappropriate structure. I suggest to answer the following aspects: - general context - novelty of the work - methodology used (describe briefly the main methods or treatments applied) - main results and related interpretations.

Reply:

Thanks for the suggestion. We follow your suggested structure to revise the Abstract as **(Lines 10-30, Pages 1-2)**: Hard-to-abate sectors annually emit about 30% of global CO₂. China's coal chemical sector is a growing carbon emitter that uses coal as both a fuel and feedstock and its greenhouse gas (GHG) emissions are hard to abate by electrification alone. Here we explore the GHG mitigation potential and costs for onsite deployment of green H₂ and O₂ and renewable electricity in China's coal chemical sector, using a life-cycle assessment and techno-economic analyses. We estimate that China's coal chemical production resulted in substantial GHG emissions of 1.1 gigaton CO₂ equivalent (GtCO₂eq) in 2020, equal to 9% of national GHG emissions. We project GHG emissions from China's coal chemical production in 2030 to be 1.3 GtCO₂eq, ~50% of which can be reduced by onsite deployment of solar or wind power-based electrolytic H₂ and O₂ to replace coal-based H₂ and air separation-based O₂ at a cost of 10 or 153 CNY/tCO₂eq, respectively. In this study, we suggest that provinces determine whether to use solar

or wind power for water electrolysis based on their lowest cost options, which collectively reduce 53% of the 2030 baseline GHG emissions at a cost of 9 CNY/tCO₂eq. We find Inner Mongolia, Shaanxi, Ningxia, and Xinjiang collectively account for 52% of total GHG mitigation with net cost reductions. These four provinces can be pioneers in deploying cost-effective green H₂ and O₂ in coal chemical production.

Comment 3:

Introduction: This section should briefly place the study in a wide context and emphasize why it is relevant carrying out the analysis. It should define the purpose of the work and its significance. In this perspective, this section is too succinct and fails to effectively point out the relevance of your contribution towards the existing literature. The authors could better introduce the topic by framing it within the energy/sustainability transition dynamics. Some literature is also required to justify the focus on green hydrogen, among the others. Please see:

<https://doi.org/10.1016/j.cogsc.2021.100506>

<https://doi.org/10.1016/j.respol.2021.104464>

<https://link.springer.com/article/10.1007/s40888-020-00206-4>

<https://doi.org/10.1016/j.eneco.2023.106642>

Reply:

Thanks for the comment. We have carefully revised the Introduction. We have investigated the suggested references and added them to enrich our literature review. We highlight some major revisions to the Introduction as follows.

For the context of literature and policies, we revised the Introduction as in **Lines 57-86, Pages 3-4**: Few studies investigated low-carbon pathways for the coal chemical sector, including product structure adjustment, conversion efficiency improvements, and carbon capture, utilization and storage^{3,6-9}. A hybrid power system integrating coal, natural gas, biomass, renewables, and nuclear was proposed as a low-carbon electricity source to produce electrolytic hydrogen for coal chemical production¹⁰. However, GHG mitigation potential and costs of deploying onsite green hydrogen for coal chemical production have not been well studied to date. In this study, we examine the benefits of deploying onsite renewable facilities nearby/within coal chemical plants to produce green H₂ and O₂ via water electrolysis. Such an approach replaces coal-based H₂ from the water-

gas shift and avoids substantial process-related CO₂ emissions (see Supplementary Figure 1). In addition, green O₂ can substitute for O₂ from coal-driven air separation and thus reduce GHG emissions from onsite fuel combustion. Onsite renewable electricity is also an alternative for grid electricity purchased by coal chemical plants, which reduces upstream GHG emissions from fossil fuels used to power the grid.

Hard-to-abate sectors account for ~30% of global annual CO₂ emissions¹¹ and transitions in their fuels and feedstocks are required for a net-zero future^{12,13}. Transitioning to a low-carbon society, such as a green hydrogen economy, is a promising pathway to climate goals^{14,15}. The chemical sector manufactures bulk materials fundamental to the economy and contributes about one eighth of global hard-to-abate emissions¹¹. Emerging technologies, especially green H₂ applications, are necessary to address these emissions from carbon-intensive chemical reactions¹⁵. The coal chemical sector is a potential large consumer for green H₂. Considering high H₂ transport costs in the near term¹⁶, onsite industrial applications are critical to large-scale development of green H₂. China has recently released strategic plans that highlight the onsite use of H₂ from renewables in the near future¹⁷. China has also initiated a series of policies to facilitate the low-carbon development of the coal chemical sector^{4,5,18}. A demonstration project within a coal chemical enterprise of Ningxia Province has deployed a utility-scale photovoltaic (PV) system to produce green H₂ for coal-to-olefin processes¹⁹.

To highlight the contribution of our work, we revised the end of the Introduction as in **Lines 90-99, Pages 4-5**: Here we explore the GHG mitigation potential and costs to decarbonize China's coal chemical sector through the onsite use of renewable electricity to produce decarbonized H₂ and O₂ and displace carbon-intensive grid-electricity. Onsite use of green H₂ in the coal chemical sector is a win-win opportunity. First, green H₂ can be used in the coal chemical sector for carbon-free feedstocks. Second, the coal chemical sector, which uses the most H₂ of any sector in China, will facilitate scale-up and cost reductions for green H₂ production. Our study provides specific implications for deploying renewables-based H₂, O₂, and electricity to produce a variety of coal based chemicals, and projects the lowest cost options for each province from now to about 2030. Our work goes beyond previous research to examine the role of green H₂ in decarbonizing the coal chemical sector nationally.

Comment 4:

Materials and methods: I found this section very important for the readability of the paper. I think the research procedure could be much more clearly described by means of a diagram also highlighting its potential and limit.

Reply:

Thanks. To clearly present our modeling procedures, we divided the Methods into three parts: GHG emission accounting, GHG mitigation modeling, and cost-benefits analyses. We describe the organization of Supplementary Tables for calculations, in the Methods, **Lines 563-567, Page 25**: All modeling procedures with data sources are provided in the Supplementary Information. We provide detailed equations and parameters in the Supplementary Tables and explain how to derive each parameter/result in the notations to each table. We arrange the Supplementary Tables in the same order as the results, namely GHG emission accounting, GHG mitigation modeling, and cost-benefit analyses.

We have added a diagram below for modeling GHG mitigation and costs as the **Figure 6** of the main text. We also have added the **Supplementary Note: Methods** (please see Supplementary Information for details) to describe the arrangement for Supplementary Tables with equations and data sources.

Figure 6. Estimation of GHG emissions and costs in the baseline and four alternative scenarios.

We also clarify the strengths and limitations of our modeling in the Methods, as in **Lines 540-562, Pages 23-25**: We integrate techno-economic analyses with a life-cycle assessment to systematically examine the GHG mitigation potential and costs of deploying onsite green H₂, green O₂, and renewable electricity in China’s coal chemical sector. Techno-economic analyses are used to examine the technical performance and cost-effectiveness of a technical process or product⁵³. A life-cycle assessment is based on a series of stages in the “cradle-to-grave” life cycle of a product or technology⁵⁴. Accordingly, we apply a broad system boundary to include onsite and upstream processes, e.g., coal gasification, the water-gas shift, air-separation for O₂, water electrolysis, upstream manufacturing of battery storage, water electrolyzers, and renewable power facilities, and upstream production of coal for chemical feedstocks and fuels.

Figure 6 presents the modeling framework for GHG mitigation and cost changes in the baseline scenario and four alternative scenarios for China's coal chemical production in 2030. We estimate 2030 baseline GHG emissions using projected coal chemical production and baseline GHG emission factors in 2030. We then quantify GHG mitigation potential of four alternative scenarios by comparing their GHG emissions to the baseline emissions. We use or estimate the 2030 projections of coal chemical production, GHG emission factors, and cost parameters based on literature, which may not fully reflect the real situation in the future. Finally, we model GHG mitigation and costs using annual averages for parameterizations at the provincial level, without considering operational parameter variations at a monthly or even finer resolution at the plant level. This limits the carbon and cost implications for coal chemical plant operations in practice.

Comment 5:

Discussions: The discussion of the results is merely descriptive and the obtained evidence is flimsy due to the fact the outcomes are not supported by an adequate discussion in light of scientific literature. Authors should discuss the results and how they can be interpreted in perspective of previous studies and their implications should be discussed in the broadest context possible.

Reply:

We expand the discussion of the Onsite Solar Electricity and Onsite Wind Electricity scenarios to obtain **provincial implications** for policymaking and practice, in **Lines 394-420, Pages 18-19**:

We explore the best options for each province to decarbonize the coal chemical sector via either solar or wind-based H₂ to achieve the maximum national GHG mitigation and minimum national costs. We use results from the Onsite Solar Electricity and Onsite Wind Electricity scenarios to optimize the combinations of provincial options to maximize national GHG mitigation and to minimize national costs. We assign each province to either solar or wind-based water electrolysis based on which technology yields larger GHG mitigation or lower costs within each province. In the maximum national GHG mitigation solution, we find a reduction of -57% (-722 MtCO₂eq) relative to 2030 baseline GHG emissions at a national annual cost addition of 26 billion CNY (36 CNY/tCO₂eq) relative to baseline costs. In this transition, 17 provinces deploy solar- and 12 provinces deploy wind-based water electrolysis (excluding Beijing and Tibet where no coal

chemicals are produced, and Hong Kong, Macau, and Taiwan where no coal chemical data are available). In the minimum national cost solution, we find a national annual cost increase of 6.3 billion CNY (9.4 CNY/tCO₂eq) in 2030 relative to baseline costs with national GHG mitigation of -53% (-665 MtCO₂eq) relative to baseline emissions. In this transition, 26 provinces deploy solar- and 3 provinces deploy wind-based water electrolysis, with the same exclusions as above. These two solutions have similar GHG mitigation (-57% vs. -53% relative to the baseline) but costs for the minimum cost solution are only 24% of those for the maximum mitigation solution. We thus suggest that provinces determine whether to use onsite solar or wind power for water electrolysis based on their lowest cost options (see Supplementary Table 23). We further find that Inner Mongolia, Shaanxi, Ningxia, and Xinjiang have much larger GHG mitigation potential than other provinces while simultaneously achieving net cost reductions due to their abundant solar energy. These four provinces collectively account for 52% of total GHG mitigation in the minimum national cost solution. For policymaking and demonstration projects, the four provinces can be pioneers in cost-effectively deploying onsite green H₂ and O₂ in coal chemical production. This clearly shows the enormous potential of decarbonizing the coal chemical sector at only a small cost increase at the national level.

We have integrated previous studies to add a discussion on battery storage versus hydrogen storage, in **Lines 433-440, Page 19**: We suggest onsite deployment of green H₂ in coal chemical plants, which can avoid costly long-distance H₂ transport³⁵. Considering battery storage is more widely used than H₂ storage³⁶, we include the GHG emissions and costs of using battery storage for renewable electricity instead of H₂ storage to reduce costs and H₂ leakage. Battery storage can help provide stable renewable electricity for water electrolysis to continuously deliver green H₂ for coal chemical production. In practice, coal chemical plants may need very short pipelines for H₂ transport within plants and small-scale H₂ storage as a back-up, which results in insignificant increases of GHG emissions and costs.

We also integrated previous studies to discuss the air quality and health co-benefits, in **Lines 441-446, Page 19**: Co-benefits for air quality and human health result from the use of green H₂ and O₂, in addition to GHG mitigation. Using green H₂ and O₂ in chemical plants can decrease onsite coal use for both feedstocks and fuels, and thus reduce air pollutant emissions from coal gasification and combustion. Avoided premature deaths from such air quality improvements can be monetized in cost-benefit analyses^{37,38} to further offset GHG mitigation costs in the SE and WE scenarios.

We combined the policy context to provide the prospects for renewable energy use in China's coal chemical sector in **Lines 466-478, Pages 20-21**: China's coal chemical plants generally have onsite captive coal power plants to generate heat and electricity for chemical production, with grid electricity as a supplementary power source⁷. In this study, we use onsite renewable electricity to electrolyze the water and to replace grid electricity purchased by coal chemical plants. The onsite coal power plants remain operational as a high-temperature heat source and as an electricity source for plant operations besides water electrolysis. China's chemical sector is expected to be included in the national carbon trading market by 2035³⁹, and high-temperature heat generation from coal is hard to replace with renewable electricity at scale in the near future. Therefore, we propose the onsite deployment of renewable electricity for water electrolysis and for replacement of grid electricity use in coal chemical plants during 2023-2035 but retain the onsite coal power plants for other plant operations. As electrification technologies advance over the next decade, we suggest that onsite deployment of renewable electricity should increasingly replace onsite heat and power generation from coal for industrial processes (such as air separation and coal gasification).

Comment 6:

Conclusions: Conclusions must also be revised according to the previous comments. In particular, they should discuss practical and policy implications as well as future lines of research. As it stands now, they fail to extract all the juice of your work.

Reply:

We have revised the conclusions to provide more implications for policymaking and future research, in **Lines 479-501, Page 21**: Onsite deployment of renewables-based electrolytic H₂ and O₂ is a feasible pathway to partially decarbonize China's coal chemical sector. We suggest that provinces determine whether to use onsite solar or wind power for water electrolysis based on their lowest cost options, which collectively reduce 53% of the 2030 baseline GHG emissions from coal chemical production at the low cost in 2030 of 9.4 CNY/tCO₂eq. We find Inner Mongolia, Shaanxi, Ningxia, and Xinjiang collectively account for 52% of total GHG mitigation that is possible with net cost reductions. These four provinces, which have extensive available land, can be pioneers in deploying cost-effective onsite green H₂ and O₂ in coal chemical production. Excess green O₂ sales can substantially reduce costs of renewables-based water electrolysis for coal chemical production.

GHG mitigation costs can be offset if the coal chemical sector is included in China's carbon trading market (the carbon price is ~50 CNY/tCO₂ in 2021⁴⁰, which makes it highly profitable to trade carbon permits when compared to the 9.4 CNY/tCO₂eq cost of mitigation in 2030). Coupling of chemical production and green hydrogen is a win-win opportunity to both scale up the deployment of green H₂ and to utilize a low-carbon feedstock for the coal-chemical sector. We plan to use plant-level operational data to extend the study of the coal chemical sector to examine the environmental co-benefits of using on-site green H₂ for air quality improvements and freshwater conservation. We will also consider the use of onsite renewable electricity with battery storage to replace captive coal power facilities in coal chemical plants when high-temperature heat generation from electricity is feasible at scale.

Comment 7:

I hope these comments might help in improving the paper and encourage the authors to move forward.

Reply:

We appreciate your valuable comments that helped improve our paper. We hope our responses address your concerns.

REVIEWERS' COMMENTS

Reviewer #1 (Remarks to the Author):

Accept

Reviewer #2 (Remarks to the Author):

The manuscript has been widely checked and amended on the basis of all the reviewers' comments. I recommend accepting it for the publication.

Reviewer #5 (Remarks to the Author):

The manuscript is much improved. I advice its publication.

Reviewer #1:

Accept

Reviewer #2:

The manuscript has been widely checked and amended on the basis of all the reviewers' comments. I recommend accepting it for the publication.

Reviewer #5:

The manuscript is much improved. I advise its publication.

Reply:

We appreciate the approval of the reviewers on our revised paper.